# Circular EZH2-encoded EZH2-92aa mediates immune evasion in glioblastoma via inhibition of surface NKG2D ligands

Jian Zhong[1,2,3,11], Xuesong Yang[1,2,3,11], Junju Chen[1,2,3,11], Kejun He[1,2,3,11], Xinya Gao[1,2,3], Xujia Wu [1,2,3], Maolei Zhang[1,2,3], Huangkai Zhou[1,2,3], Feizhe Xiao[4], Lele An[5,6], Xiuxing Wang [7,8,9,10] ✉, Yu Shi [5,6] ✉ & Nu Zhang [1,2,3] ✉

Glioblastoma (GBM) is a highly aggressive primary brain tumour and is resistant to nearly all available treatments, including natural killer (NK) cell immunotherapy. However, the factors mediating NK cell evasion in GBM remain largely unclear. Here, we report that EZH2-92aa, a protein encoded by circular EZH2, is overexpressed in GBM and induces the immune evasion of GBM stem cells (GSCs) from NK cells. Positively regulated by DEAD-box helicase 3 (DDX3), EZH2-92aa directly binds the major histocompatibility complex class I polypeptide-related sequence A/B (MICA/B) promoters and represses their transcription; it also indirectly represses UL16-binding protein (ULBP) transcription by stabilizing EZH2. The downregulation of NK group 2D ligands (NKG2DLs, including MICA/B and ULBPs) in GSCs mediates NK cell resistance. Moreover, stable EZH2-92aa knockdown enhances NK cell-mediated GSC eradication in vitro and in vivo and synergizes with anti-PD1 therapy. Our results highlight the immunosuppressive function of EZH2-92aa in inhibiting the NK cell response in GBM and the clinical potential of targeting EZH2-92aa for NK-cell-directed immune therapy.

Glioblastoma (GBM) is the most lethal primary brain tumour in adults. Despite aggressive treatments, including surgical resection, radiotherapy and chemotherapy, the outcomes of patients with GBM are dismal, with a median survival time of <2 years and a 5-year survival rate of only 5.8%[1]. While recent advances in cancer immunotherapy have improved patient outcomes in certain types of cancer, the immunosuppressive tumour microenvironment (TME) of GBM poses a major therapeutic challenge in malignant brain cancers[2,3].

Natural killer (NK) cells are large granular lymphocytes that can spontaneously lyse malignant cells, including GBM stem cells (GSCs)[4–6], suggesting their distinct advantages over T cells for therapeutic approaches[7–9]. However, the results of clinical trials

[1]Department of Neurosurgery, The First Affiliated Hospital of Sun Yat-sen University, Guangzhou, Guangdong 510080, China. [2]Guangdong Provincial Key Laboratory of Brain Function and Disease, Guangzhou, Guangdong 510080, China. [3]Institute of Precision Medicine, The First Affiliated Hospital of Sun Yat-sen University, Guangzhou, Guangdong 510080, China. [4]Department of Scientific Research Section, The First Affiliated Hospital of Sun Yat-sen University, Guangzhou, Guangdong 510080, China. [5]Institute of Pathology and Southwest Cancer Centre, Southwest Hospital, Third Military Medical University (Army Medical University), Chongqing 400038, China. [6]Key Laboratory of Tumour Immunopathology of the Ministry of Education of China, Southwest Hospital, Third Military Medical University (Army Medical University), Chongqing 400038, China. [7]National Health Commission Key Laboratory of Antibody Techniques, School of Basic Medical Sciences, Nanjing Medical University, Nanjing, Jiangsu 211166, China. [8]Department of Cell Biology, School of Basic Medical Sciences, Nanjing Medical University, Nanjing, Jiangsu 211166, China. [9]Jiangsu Provincial Key Laboratory of Human Functional Genomics, School of Basic Medical Sciences, Nanjing Medical University, Nanjing, Jiangsu 211166, China. [10]Institute for Brain Tumors, Jiangsu Key Lab of Cancer Biomarkers, Prevention and Treatment, Collaborative Innovation Center for Personalized Cancer Medicine, Nanjing Medical University, Nanjing, Jiangsu 211166, China. [11]These authors contributed equally: Jian Zhong, Xuesong Yang, Junju Chen, Kejun He. ✉e-mail: drxiuxingwang@163.com; drshiyu@126.com; zhangnu2@mail.sysu.edu.cn

using NK cells to target GBM were largely unsatisfactory[10], prompting studies on the mechanism by which GSCs evade NK cell surveillance and eradication. NK cell cytotoxicity is facilitated by an array of activating receptors[11]. Disruption of receptor–ligand interactions between NK cells and GSCs is the primary determinant of NK cell resistance, as downregulation of NK group 2D (NKG2D) ligands is frequently observed in GBM[12,13]. However, the factors that determine the aberrant expression of NKG2D ligands remain largely unknown.

Recently, we reported the hidden functions of circular RNA (circRNA)-encoded proteins in GBM[14,15]. CircRNA dysregulation is frequently observed in cancers, leading to the hypothesis that imbalanced expression of circRNA-encoded proteins can contribute to tumorigenesis and tumour progression[16]. However, very little is known regarding whether these proteins participate in immunosuppressive signalling, especially NK cell resistance, in GSCs.

In this work, we find that circular EZH2 (circEZH2) plays a critical role in suppressing NK cell cytotoxicity in GBM. CircEZH2-encoded EZH2-92aa is overexpressed in clinical GBM tumours and decreases susceptibility to NK cell cytotoxicity by suppressing the expression of NKG2D ligands, suggesting that NK cell-based therapies can be combined with EZH2-92aa targeting strategies to improve clinical outcomes of this lethal tumour.

## Results

### CircEZH2 is highly expressed in GBM and negatively correlated with the NK cell activation signature

GBM tumours are frequently infiltrated by NK cells[17,18], and greater NK cell infiltration predicts longer overall survival (OS) in patients with GBM in The Cancer Genome Atlas (TCGA) (Supplementary Fig. 1a). However, infiltrated NK cells are also actively suppressed by tumour cells or the tumour microenvironment, which limits the cytolytic activity of NK cells against GBM cells[12,13,19]. We first performed RNA-seq analyses of tumour specimens (T) and paired normal brain tissues (N) from 12 patients with GBM (accession ID: PRJNA525736) to investigate potential circRNAs related to NK cell functions. A total of 2289 differentially expressed circRNAs were identified, 984 of which were upregulated in tumour samples (Fig. 1a, Supplementary Data 1). Recent studies have revealed the importance of small open reading frames (ORFs) in circRNAs in tumorigenesis[14,20]. To investigate the potential coding capability of the upregulated circRNAs, we leveraged a previously reported scoring approach[21] to annotate ORFs in their sequences. Through this approach, we identified 473 circRNAs that may encode uncharacterized peptides with junction-spanning ORFs (circORFs). Compared with their linear cognates (main ORFs, mORFs), circORFs only encoded significantly shorter peptides, with a peak length distribution of less than 100 amino acids (aa) (Fig. 1b). A total of 164 circORFs among those 473 candidates (34.7%) were shorter than 100 aa (Supplementary Data 2). We next focused on circRNAs that both encoded putative small peptides shorter than 100 aa and were significantly overexpressed in tumour samples (Fig. 1c). The top 10 candidates were thus identified (Fig. 1d).

To narrow the list of NK cell signalling-related candidates, we subsequently performed a gene set enrichment analysis (GSEA) and found that only circEZH2, which ranked 6th among the overexpressed circORFs in GBM, was negatively correlated with 'Natural Killer Cell Mediated Cytotoxicity' (Fig. 1e, Supplementary Data 3). Moreover, the expression of the identified NK activation gene set[7], including *EGR2*, *EGR3*, *IFNG*, *TNF* and *GZMB* (Supplementary Table 1), was negatively correlated with circEZH2 expression ($P = 0.012$) (Fig. 1f). These results suggested that circEZH2 is highly expressed in GBM and may be involved in suppressing NK cell functions. We then focused on circEZH2 for further investigation.

### Characterization of circEZH2 in GBM

The 253-nt circEZH2 is formed by circularization of exons 2 and 3 of the *EZH2* gene (Fig. 1g). The predicted backsplice junction was validated via Sanger sequencing and was consistent with the circBase database (ID: hsa_circ_0006357, Fig. 1h). In contrast to linear EZH2 mRNA, circEZH2 was resistant to RNase R digestion and had a longer half-life (Supplementary Fig. 1b, c). CircEZH2 was successfully amplified only by random primers, while the linear EZH2 mRNA was amplified by both random and oligo (dT) primers in MES28 patient-derived GSCs (Supplementary Fig. 1d). By performing immunofluorescence (IF) and cell fractionation qPCR, we found that similar to EZH2 mRNA, circEZH2 was localized mainly in the cytoplasm (Fig. 1i and Supplementary Fig. 1e). We also designed two shRNAs (shRNA-1 and shRNA-2) that specifically targeted the circEZH2 backspliced junction, as shown in Fig. 1i. By using a junction probe, we confirmed that these shRNAs successfully knocked down circEZH2 in MES28 GSCs and that the circEZH2 overexpression (OV) plasmid successfully upregulated circEZH2 (Fig. 1i). In addition, circEZH2 was overexpressed in several patient-derived GSC lines, including MES28 and GSC23, compared with normal human astrocytes (NHAs) (Fig. 1j). In a cohort of clinical samples (63 randomly selected paired high-grade glioma samples), the levels of circEZH2 were significantly higher in the tumour samples than in the paired adjacent brain tissues ($P < 0.01$) (Fig. 1k, l). Based on these findings, circEZH2 is a potential upregulated immunosuppressive circRNA in GBM.

### CircEZH2 encodes the peptide EZH2-92aa

Translatable circRNAs and their protein/peptide products have recently been reported[21,22]. Specifically, we showed that ORFs spanning more than 360 degrees in circRNAs can generate unique molecular targets in cancers[14]. In circEZH2, a similar ORF was identified that potentially generated a protein (named EZH2-92aa hereafter) with a unique C-terminus through a frameshift in the second round of translation (10 extra unique aa) (Fig. 2a). A conserved ORF was also identified in murine circEZH2 (circBase ID: mmu_circ_0001471, Supplementary Fig. 2a, b). To validate the coding potential of circEZH2, we transfected circEZH2 into 293T cells and performed quantitative polymerase chain reaction (qPCR) in the nonribosomal, 40–80S ribosomal and polysomal fractions[22]. Like linear EZH2 mRNA (positive control), circEZH2 was detected in the 40–80S ribosome and polysome fractions, which implied its translational potential. In contrast, most circHIPK3 (negative control) transcripts were distributed in the nonribosomal fraction (Fig. 2b). Similar to most translatable circRNAs, translation of the ORF in circEZH2 was driven by an internal ribosome entry site (IRES), whose activity was validated by a circular vector-based luciferase assay (Supplementary Fig. 2c). Subsequently, we generated a specific antibody that targeted the unique C-terminal amino acid sequence of human and murine EZH2-92aa (LRGTRENNHGPDWEEI) (Fig. 2a and Supplementary Fig. 2b). This antibody recognized endogenous EZH2-92aa encoded by circEZH2 as well as linearized EZH2-92aa-3×Flag in transfected 293T cells. When the start codon 'ATG' in circEZH2 was deleted, overexpression of EZH2-92aa was not detected (Fig. 2c, d). Furthermore, overexpressed EZH2-92aa in 293T cells and endogenous EZH2-92aa in MES28 GSCs and the murine glioma cell line GL261 were identified by mass spectrometry (MS) (Fig. 2e and Supplementary Fig. 2d, e). The MS-identified unique C-terminal amino acid sequences strongly supported the existence of EZH2-92aa. EZH2-92aa expression was substantially higher in GSCs than in NHAs, consistent with the circEZH2 expression level (Fig. 1j and Fig. 2f, top panel). Due to the unique C-terminus that was identical to the sequence in humans, murine EZH2-92aa can also be detected by this antibody, and its stable knockdown in GL261 cells by junction-specific shRNAs was verified (Supplementary Fig. 2f). In 7 randomly selected GBM samples and paired normal tissues, EZH2-92aa was expressed at high levels in the tumour tissues (Fig. 2f, bottom panel).

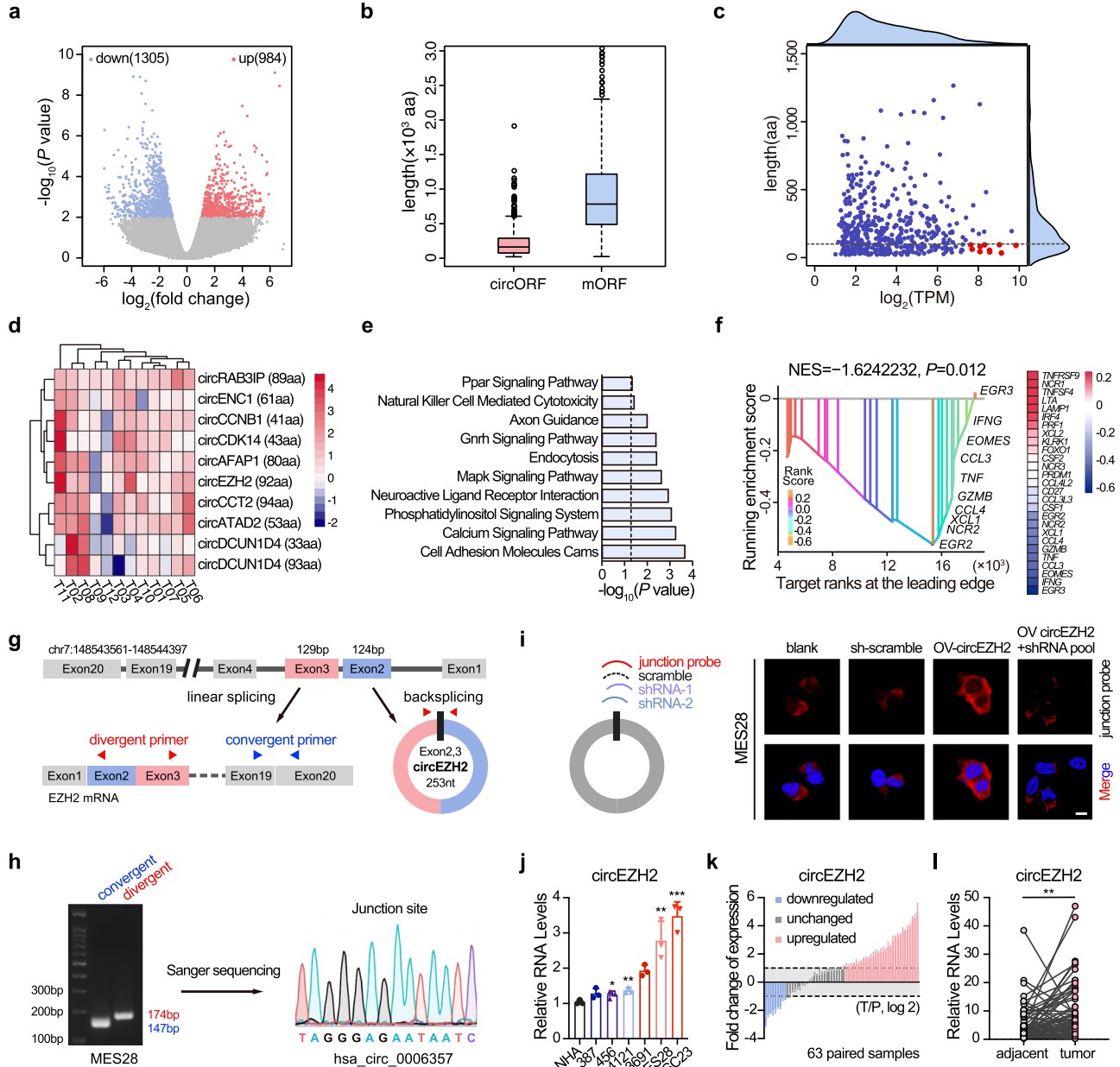

**Fig. 1 | CircEZH2 is highly expressed in GBM and negatively correlated with the NK cell activation signature. a** Volcano plot of differentially expressed circRNAs in 12 paired human GBM samples. **b** Box plot of the length distribution of circORFs in tumour-upregulated circRNAs and the corresponding mORFs of the source genes. Data are presented as boxes containing the median (centre line), the first and third quartiles (box limits). The whiskers indicate the maxima and minima. **c** Expression distribution (x-axis) and length distribution (y-axis) of circORFs in tumour-upregulated circRNAs. The red dots represent the 10 candidate circRNAs. **d** Heatmap of the fold change values of the 10 circRNAs in (**c**). **e** Pathways negatively correlated with circEZH2 in GSEA. **f** Correlation analysis between circEZH2 and the functional NK cell-associated marker gene set (27 genes) in GSEA. The heatmap on the right shows the Pearson correlation coefficient between each marker and circEZH2. The vertical bars in the figure indicate the ranks and enrichment scores of the 27 genes as determined by GSEA. Ten core enriched genes are marked in the plot. **g** Illustration of the annotated genomic region of EZH2, the putative different RNA splicing forms. Convergent and divergent primers were designed to amplify the linear- or back-spliced products. **h** Left, PCR analysis using the indicated primers. Right, subsequent Sanger sequencing identified the junction sequence of circEZH2 in MES28 cells. **i** Left, illustration of the circEZH2 junction-specific FISH probe and circEZH2 shRNA target site. Right, FISH was performed to identify the subcellular localization of circEZH2. CircEZH2 OV plasmids and shRNA were used independently or in combination to verify the specificity of these probes. Scale bar, 10 μm. **j** Relative circEZH2 RNA levels in NHAs and GSC lines. 456 vs NHA, $P = 0.0389$; 4121 vs NHA, $P = 0.0056$; MES28 vs NHA, $P = 0.0056$; GSC23 vs NHA, $P = 0.0005$. **k** Fold change in circEZH2 expression in tumour specimens and paired adjacent brain tissues in a cohort of high-grade glioma patients ($n = 63$). **l** Relative circEZH2 expression levels in the same cohort ($n = 63$). Two-sided paired $t$ test, $P = 0.0013$. The data in (**h**)–(**l**) are pooled from three independent experiments. The data are presented as the mean ± SD. Unpaired two-tailed Student's $t$ test was used to determine the significance of differences between the indicated groups where applicable. *$P < 0.05$; **$P < 0.01$; ***$P < 0.001$. Source data are provided as a Source Data file.

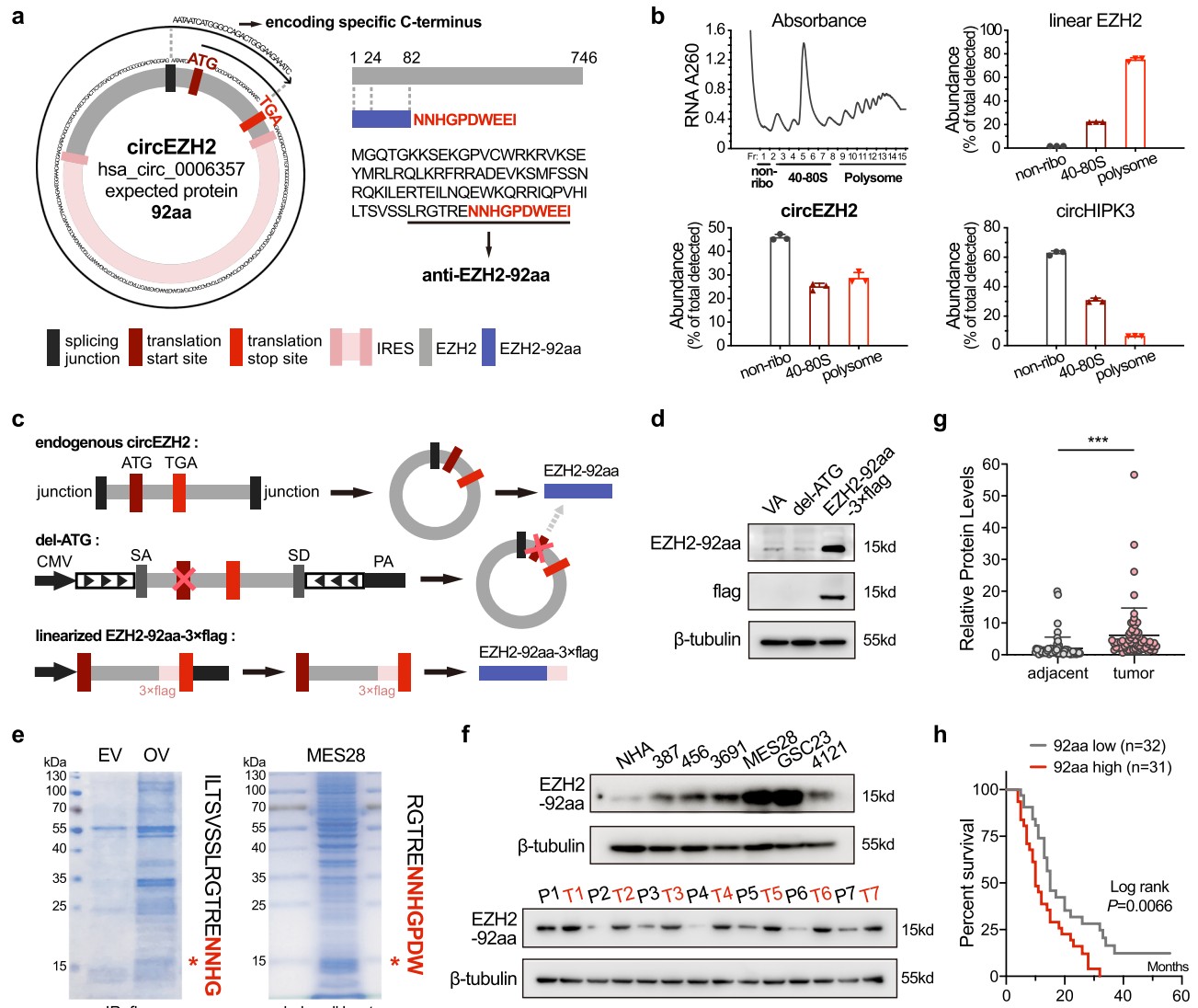

**Fig. 2 | CircEZH2 encodes the peptide EZH2-92aa. a** Illustration of the circEZH2-encoded peptide EZH2-92aa. The unique C-terminus of EZH2-92aa (in red) is produced by an ORF spanning more than 360° in circEZH2. A custom antibody against the indicated unique C-terminal sequence was produced. **b** 293T cells transfected with the circEZH2 plasmid were subjected to polysome profiling. CircEZH2 was detected by qPCR in the indicated fractions. CircHIPK3 and EZH2 served as the negative and positive controls, respectively. **c** Illustration of endogenous circEZH2, the del-ATG circEZH2 construct and the linearized EZH2-92aa-3×Flag construct. **d** Immunoblot of cells overexpressing the above constructs using the custom anti-EZH2-92aa and anti-flag antibodies. VA, vector alone. **e** Identification of the unique C-terminal peptide sequence of EZH2-92aa in 293T cells overexpressing circEZH2 and in MES28 GSCs by MS. **f** Top, EZH2-92aa expression levels were measured in NHAs and several established GSC lines. Bottom, immunoblot of EZH2-92aa in seven randomly selected paired GBM samples using the custom anti-EZH2-92aa antibody. **g** Semiquantitative analysis of the EZH2-92aa expression level based on greyscale analysis in the aforementioned cohort of 63 high-grade glioma samples. Two-sided paired $t$ test, $P = 3.14e−05$. **h** Survival analysis of patients stratified by EZH2-92aa expression (with the median expression score as the cut-off value) in the same cohort. Log-rank test, $P = 0.0066$. The data in (**b**), (**d**), (**e**) and (**f**) are pooled from three independent experiments and are presented as the mean ± SD. Source data are provided as a Source data file.

Semiquantitative scoring of the western blot analysis indicated that EZH2-92aa was preferentially expressed in tumour tissues in our in-house cohort of 63 high-grade glioma samples (Fig. 2g). We defined the EZH2-92aa expression level as 'EZH2-92aa high' or 'EZH2-92aa low' (with the median expression level defined as the cut-off) based on semiquantitative western blot analysis and found that higher EZH2-92aa expression was negatively correlated with OS (Fig. 2h). Moreover, a multivariate Cox regression analysis was subsequently performed on the same cohort and demonstrated that EZH2-92aa expression, age, IDH1 mutation and 1p19q status were all significantly associated with OS (Supplementary Fig. 2g). These data indicated that EZH2-92aa may be an independent prognostic marker in high-grade glioma.

## EZH2-92aa translation is regulated by DEAD-box helicase 3 (DDX3)

CircRNA translation depends primarily on IRESs or N6-methyladenosine ($m^6A$)[23,24]. Whether other factors enhance or facilitate circRNA translation is largely unknown. Specifically, the global regulators of translation, DEAD-box family RNA-binding proteins, have been reported to be involved in IRES-driven translation in viruses[25,26]. In addition, a previous study indicated that circRNAs can interact with DDX3 to transactivate YY1-induced transcriptional alteration of downstream genes[27]. By performing RNA immunoprecipitation (RIP) and subsequent reverse transcription-qPCR (RT-qPCR) analysis with specific primers, we found that DDX3 interacted with the IRES sequence of circEZH2 in

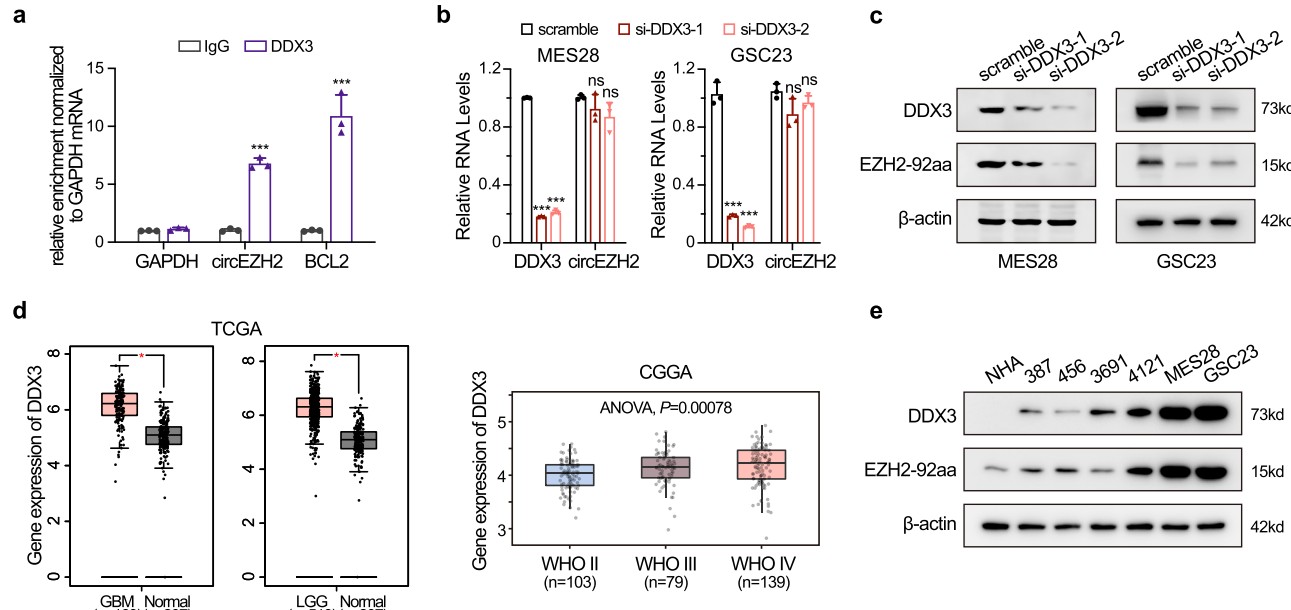

**Fig. 3 | EZH2-92aa translation is regulated by DDX3. a** MES28 GSCs were subjected to RNA immunoprecipitation using an anti-DDX3 antibody or IgG (negative control). Immunoprecipitates were analysed by RT-qPCR with specific primers for the circEZH2 IRES sequence ($P = 3.35e−05$), GAPDH (negative control, $P = 0.0636$) and BCL2 (positive control, $P = 7.09e−04$). **b** DDX3 and circEZH2 RNA levels in MES28 and GSC23 cells transfected with DDX3 siRNAs. MES28, si-DDX3-1, $P = 8.66e−10$, si-DDX3-2, $P = 9.31e−08$; GSC23, si-DDX3-1, $P = 5.77e−05$, si-DDX3-2, $P = 4.06e−05$. **c** Protein levels of DDX3 and EZH2-92aa in MES28 and GSC23 cells transfected with DDX3 siRNAs. **d** Expression of DDX3 in GBM datasets from TCGA (GBM and LGG dataset, plotted by GEPIA2, [http://gepia.cancer-pku.cn/]) and CGGA (dataset

mRNAseq_325, [http://www.cgga.org.cn/index.jsp]). Data are presented as boxes containing the median (centre line), the first and third quartiles (box limits). The whiskers indicate the maxima and minima. One-way ANOVA test. For TCGA dataset, *$P < 0.01$. For CGGA dataset, $P = 0.00078$. **e** Protein levels of DDX3 and EZH2-92aa in NHAs and several patient-derived GSC lines. The data in (**a**), (**b**), (**c**) and (**e**) are pooled from three independent experiments. The data are presented as the mean ± SD. Unpaired two-tailed Student's *t* test was used to determine the significance of differences between the indicated groups. *$P < 0.05$; **$P < 0.01$; ***$P < 0.001$. Source data are provided as a Source data file.

MES28 GSCs (Fig. 3a). To investigate whether DDX3 is involved in circEZH2 translation, we knocked down DDX3 by two specific siRNAs in MES28 GSCs. DDX3 knockdown (KD) reduced DDX3 mRNA expression without affecting the circEZH2 level (Fig. 3b). In contrast, EZH2-92aa expression was substantially inhibited after DDX3 KD, indicating that circEZH2 translation depended on DDX3 (Fig. 3c). DDX3 is overexpressed in GBMs compared with normal brain tissues, as supported by the analysis of TCGA and the Chinese Glioma Genome Atlas (CGGA) databases (Fig. 3d). In patient-derived GSCs, the EZH2-92aa level was positively correlated with the DDX3 protein level (Fig. 3e). Therefore, our data suggested that the RNA-binding protein DDX3 is a potential enhancer of EZH2-92aa translation in GBM.

### EZH2-92aa inhibits NK cell cytotoxicity

To investigate the potential functions of EZH2-92aa, we first performed RNA-seq in circEZH2 stable KD MES28 GSCs and control GSCs (expressing scrambled shRNA). GSEA revealed that circEZH2 expression was negatively correlated with the expression of an established set of NK cell-activating factors (Fig. 4a, Supplementary Table 2), consistent with previous sequencing results. Given the correlation of circEZH2 with NK cell activity, we employed the NK-92MI cell line and primary NK cells sorted from peripheral blood mononuclear cells (PBMCs) from two different donors (Supplementary Fig. 3a) and performed cell-based cytotoxicity assays to assess NK cell-mediated cytotoxicity towards cancer cells. First, we measured lactate dehydrogenase (LDH) activity in culture medium after coculturing NK cells with GSCs with the indicated modifications. Stable KD of circEZH2 enhanced the cytotoxicity of NK cells (effector, E) in both MES28 and GSC23 GSCs (target, T) compared with the corresponding control cells at different T:E ratios (Fig. 4b,

left panel and Supplementary Fig. 3b, c). Conversely, stable OV of circEZH2 or the EZH2-92aa ORF promoted the resistance of GSC456 cells to NK cells. However, OV of mutated circEZH2 (del-ATG), which did not produce EZH2-92aa, failed to enhance the resistance of GSCs to NK cells (Fig. 4b, right panel and Supplementary Fig. 3b). These results suggested that EZH2-92aa but not circEZH2 exerts the main biological effects.

To validate the above findings, we next performed a calcein AM dye-based fluorescent imaging method to measure NK cell-mediated cytotoxicity. MES28 and GSC23 were labelled with calcein AM dye and incubated with NK cells, and the resulting NK cell-mediated cytotoxicity was quantified using fluorescence imaging. Consistent with the results of the LDH-based cytotoxicity assay, the calcein AM dye-based fluorescent assay indicated that EZH2-92aa expression was positively correlated with NK cell resistance in those GSC lines (Fig. 4c, d and Supplementary Fig. 3d–g). We also characterized the markers associated with degranulation, cytokine production and activation of NK-92MI cells. Stable KD of circEZH2 in MES28 and GSC23 cells promoted the expression of CD107a, Granzyme B, Perforin, IFN-γ and TNF-α in cocultured NK cells (Fig. 4e, f), while stable OV of either circEZH2 or EZH2-92aa suppressed the expression of the aforementioned markers (Supplementary Fig. 3h), suggesting enhanced NK cell degranulation and cytokine production upon circEZH2 inhibition. Consistent with these results, stable OV of del-ATG circEZH2 in GSCs did not inhibit the expression of those proteins in cocultured NK cells, providing further supporting evidence that EZH2-92aa but not circEZH2 inhibits NK cell function in vitro (Supplementary Fig. 3h). Notably, the knockdown of DDX3, which is the potential positive regulator of EZH2-92aa translation, also increased NK cytotoxicity, according to the results of LDH and calcein AM assays (Supplementary Fig. 3i–l).

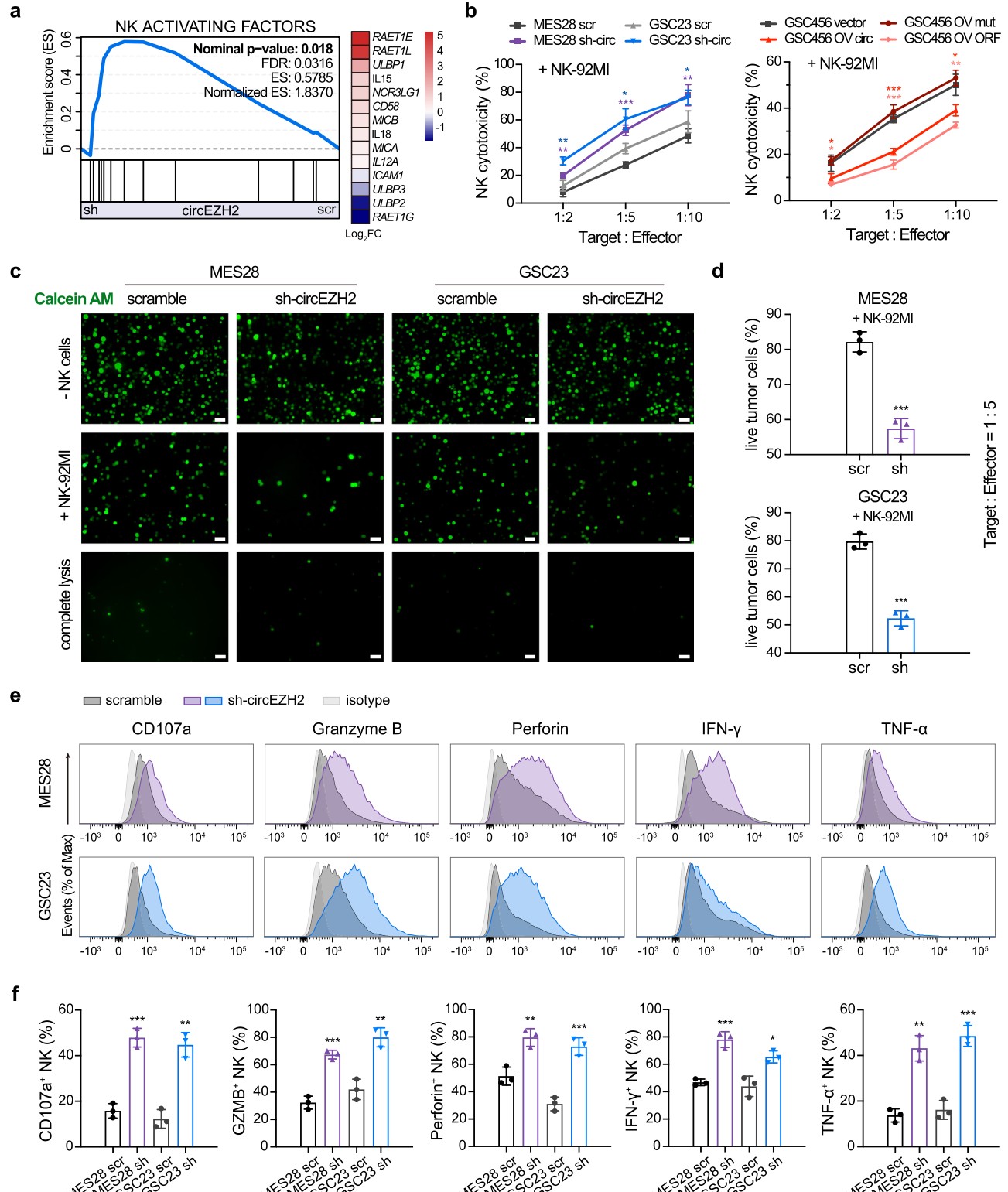

**EZH2-92aa directly represses the transcription of the NKG2D ligands major histocompatibility complex class I polypeptide-related sequence A/B (MICA/B)**

Cancer cells may evade NK cell-mediated cytotoxicity by dysregulating MHC-I expression[28], increasing the expression of dysfunction markers (including PD-L1, TGF-β, etc.)[11], disturbing natural cytotoxicity receptor (NCR)-mediated activation[29], or repressing the expression of NKG2D ligands (NKG2DLs, including ULBPs and MICA/B)[12,19,30,31]. To elucidate the mechanism by which EZH2-92aa mediates

NK cell tolerance, we next determined the subcellular localization of EZH2-92aa. Immunofluorescence imaging and subcellular fractionation followed by western blotting showed that EZH2-92aa was localized primarily in the nucleus, although some EZH2-92aa was localized in the cytoplasm (Fig. 5a). We next assessed the levels of MHC-I and a series of NK cell dysfunction markers and activation factors (ligands for NCR and NKG2D) in circEZH2 stable KD MES28 cells. Compared with GSCs transfected with the scrambled shRNA, circEZH2 KD GSCs did not exhibit alterations in NK cell dysfunction

**Fig. 4 | EZH2-92aa inhibits NK cell cytotoxicity. a** Left, GSEA of an established set of NK cell-activating factors in circEZH2 stable KD and control MES28 GSCs. Right, heatmap of the fold change values of NK cell-activating factors in the established gene set. scr, scrambled shRNA control; sh, sh-circEZH2 RNA. **b** GSCs with stable circEZH2 KD or OV were cocultured with NK cells at different T:E ratios for 2 h. LDH activity in the supernatant was measured to calculate NK cell cytotoxicity. scr, scrambled shRNA control; sh-circ, sh-circEZH2 RNA; OV mut, overexpression of mutated circEZH2 with deletion of ATG; OV circ, overexpression of circEZH2; OV ORF, overexpression of the linearized circEZH2 ORF. MES28, scr vs sh-circ, 1:2 $P = 0.0050$, 1:5 $P = 0.0006$, 1:10 $P = 0.0051$; GSC23, scr vs sh-circ, 1:2 $P = 0.0029$, 1:5 $P = 0.0120$, 1:10 $P = 0.0319$; GSC456, vector vs OV circ, 1:2 $P = 0.0494$, 1:5 $P = 0.0003$, 1:10 $P = 0.0207$; GSC456, vector vs OV ORF, 1:2 $P = 0.0124$, 1:5 $P = 0.0002$, 1:10 $P = 0.0031$. **c** GSCs were stained with calcein AM and seeded with NK cells at a T:E ratio of 1:5. Images of fluorescent cells (live cells) were acquired after incubation for 2 h. GSCs cultured without NK cells (-NK cells) and completely lysed GSCs served as the negative and positive controls, respectively. Scale bar, 50 μm. **d** Quantification of remaining live GSCs in the calcein AM cytotoxicity assay. scr, scrambled shRNA control; sh, sh-circEZH2 RNA. scr vs sh, MES28 $P = 4.59e-04$, GSC23 $P = 2.44e-04$. **e, f** Representative histogram (**e**) and quantification (**f**) of the expression of the indicated molecules in NK cells after incubation with circEZH2 stable KD or control GSCs. scr, scrambled shRNA control; sh, sh-circEZH2 RNA. scr vs sh, MES28, CD107a $P = 0.0004$, GZMB $P = 0.0004$, Perforin $P = 0.0062$, IFN-γ $P = 0.0010$, TNF-α $P = 0.0012$; GSC23, CD107a $P = 0.0011$, GZMB $P = 0.0030$, Perforin $P = 0.0009$, IFN-γ $P = 0.0125$, TNF-α $P = 0.0008$. The data are presented as the mean ± SD of three independent experiments. Unpaired two-tailed Student's $t$ test was used to determine the significance of differences between the indicated groups. *$P < 0.05$; **$P < 0.01$; ***$P < 0.001$. Source data are provided as a Source data file.

marker, NCR ligand or MHC-I expression levels (Supplementary Fig. 4a–c). In contrast, NKG2D ligands, including MICA and MICB, were upregulated at the transcriptional level after circEZH2 KD (Fig. 5b and Supplementary Fig. 4d). A nuclear run-on (NRO) assay also revealed increased or reduced nascent transcription of MICA/B in circEZH2 KD or EZH2-92aa stable OV GSCs, respectively (Supplementary Fig. 4e). The increased expression of MICA/B proteins was confirmed by western blot and flow cytometry analyses (Fig. 5b and Supplementary Fig. 4d, f). Knockdown of DDX3, which inhibited the translation of EZH2-92aa, also restored the expression of MICA/B in those GSCs (Supplementary Fig. 4g). Considering the nuclear localization of EZH2-92aa, we asked whether it directly dysregulates the transcription of these NKG2D ligands. After cloning the promoters of these NKG2D ligands into a luciferase reporter vector and cotransfecting them with EZH2-92aa, we found that EZH2-92aa repressed MICA/B promoter activity in a dose-dependent manner (Fig. 5c, d). Using truncated MICA/B promoters, we observed that the −750 to −500 bp segment of the MICA promoter and the −600 to −300 bp segment of the MICB promoter were critical for EZH2-92aa-mediated repression (Fig. 5e, f).

We next performed a chromatin immunoprecipitation (ChIP) assay in EZH2-92aa-3×Flag-transfected MES28 GSCs with specifically designed primers targeting the MICA/B promoter regions[30]. EZH2-92aa-3×Flag bound to the MICA/B promoters, further indicating that EZH2-92aa may directly repress MICA/B transcription (Fig. 5g, h). We also employed the custom anti-EZH2-92aa antibody and validated the endogenous binding of EZH2-92aa to MICA/B promoters (Supplementary Fig. 4h). To investigate whether EZH2-92aa directly interacts with the MICA/B promoters and narrow down the potential binding sites, we designed a set of probes targeting the abovementioned precipitated regions (Fig. 5i, j and Supplementary Fig. 4i) and performed an electrophoretic mobility shift assay (EMSA). EZH2-92aa-3×Flag interacted with MICA probe 3 and its truncated forms probe 4 and probe 6 (−646 to −627 bp) but not with probes 1 and 2 in MES28 GSCs, indicating a direct interaction (Fig. 5i). Similarly, MICB probe 1 and its truncated form probe 6 (−487 to −472 bp) successfully interacted with EZH2-92aa (Fig. 5j). The interaction between endogenous EZH2-92aa and MICA/B probes was also verified by using nuclear extracts of MES28 and GSC23 GSCs (Supplementary Fig. 4j). By analysing the above shifted probes, we identified a shared sequence, 'GGAGAA' (Supplementary Fig. 4i). When GGAGAA was mutated to GGCTAA, the mutant (Mut) competitor failed to block the interaction. Further addition of an anti-Flag antibody resulted in the formation of a supershifted band, validating the specificity of the interaction (Fig. 5k, l). Similarly, when GGAGAA was mutated to GGCTAA in the MICA/B promoters, EZH2-92aa-3×Flag did not induce repression in the luciferase assay (Fig. 5m, n). The above evidence collectively demonstrated that EZH2-92aa can transcriptionally repress MICA/B by directly binding to their promoters.

## EZH2-92aa indirectly represses the NKG2D ligand ULBP1 by stabilizing EZH2

We next investigated whether EZH2-92aa was involved in dysregulating ULBPs, another set of NKG2D ligands. EZH2-92aa KD in MES28 GSCs increased ULBP1/4/6 mRNA expression, consistent with the GSEA results as validated by qPCR (Figs. 4a and 6a). However, we failed to immunoprecipitate the promoter of ULBP1/4/6 in the ChIP-qPCR assays, suggesting that EZH2-92aa may suppress ULBP1/4/6 expression via other indirect mechanisms (Supplementary Fig. 5a). Interestingly, EZH2 was reported to repress ULBPs and induce NK cell evasion by trimethylating H3K27 in their promoter regions[32]. Given that EZH2-92aa is also localized in the cytoplasm, we then explored whether EZH2-92aa can protect EZH2 from degradation, as we previously reported[33,34].

Knockdown of EZH2-92aa in MES28 and GSC23 GSCs by siRNAs reduced the EZH2 protein level without changing the EZH2 mRNA level (Fig. 6b). Consistent with this finding, the H3K27me3 level was also reduced (Fig. 6b, right panel). In addition, EZH2-92aa KD drastically reduced the half-life of EZH2 compared with that in scrambled control cells (Fig. 6c). The addition of MG132, a proteasome inhibitor, restored the EZH2 protein level in both MES28 and GSC23 cells with EZH2-92aa KD (Fig. 6d). Further investigation revealed that FBXW7, a known E3 ligase that directly enhances EZH2 degradation[35], interacted with EZH2-92aa in MES28 GSCs (Fig. 6e). FBXW7 transfection reduced the EZH2 protein level, while this effect was reversed by EZH2-92aa OV, further indicating that EZH2-92aa protects EZH2 from degradation mediated by FBXW7 (Fig. 6f). We next performed ChIP-qPCR in EZH2-92aa stable KD MES28 GSCs. An evaluation of H3K27me3 by ChIP-qPCR demonstrated that EZH2-92aa KD reduced the level of H3K27me3 bound to the ULBP1 promoter in MES28 GSCs, indicating that EZH2-92aa indirectly represses NKG2D ligand transcription via EZH2 stabilization (Fig. 6g). In circEZH2 stable KD MES28 and GSC23 cells, EZH2 OV reduced the ULBP1 mRNA level (Fig. 6h). These data suggested that EZH2-92aa can repress ULBP1 transcription by stabilizing EZH2. To further determine EZH2-92aa regulated NK cell activation via the NKG2DL-NKG2D axis, we also applied an NKG2D blocking antibody and repeated the NK cytotoxicity assays. The addition of the NKG2D blocking antibody significantly reversed the effect of EZH2-92aa KD, thus further suggesting EZH2-92aa mainly impaired NK cell activation by interfering with NKG2DL expression (Fig. 6i, j and Supplementary Fig. 5b–d).

## Inhibition of EZH2-92aa sensitizes GSCs to NK cell cytotoxicity and synergizes with immune checkpoint blockade

Given that EZH2-92aa expression endowed GSCs with the ability to evade NK cell cytotoxicity, we next investigated whether EZH2-92aa inhibition can enhance NK cell-induced tumour eradication in vivo. MES28 and GSC23 GSCs with stable circEZH2 KD were implanted into the brains of immunocompromised mice (B-NDG mice) (Fig. 7a). Stable KD of circEZH2 reduced the tumour volume and prolonged the OS of

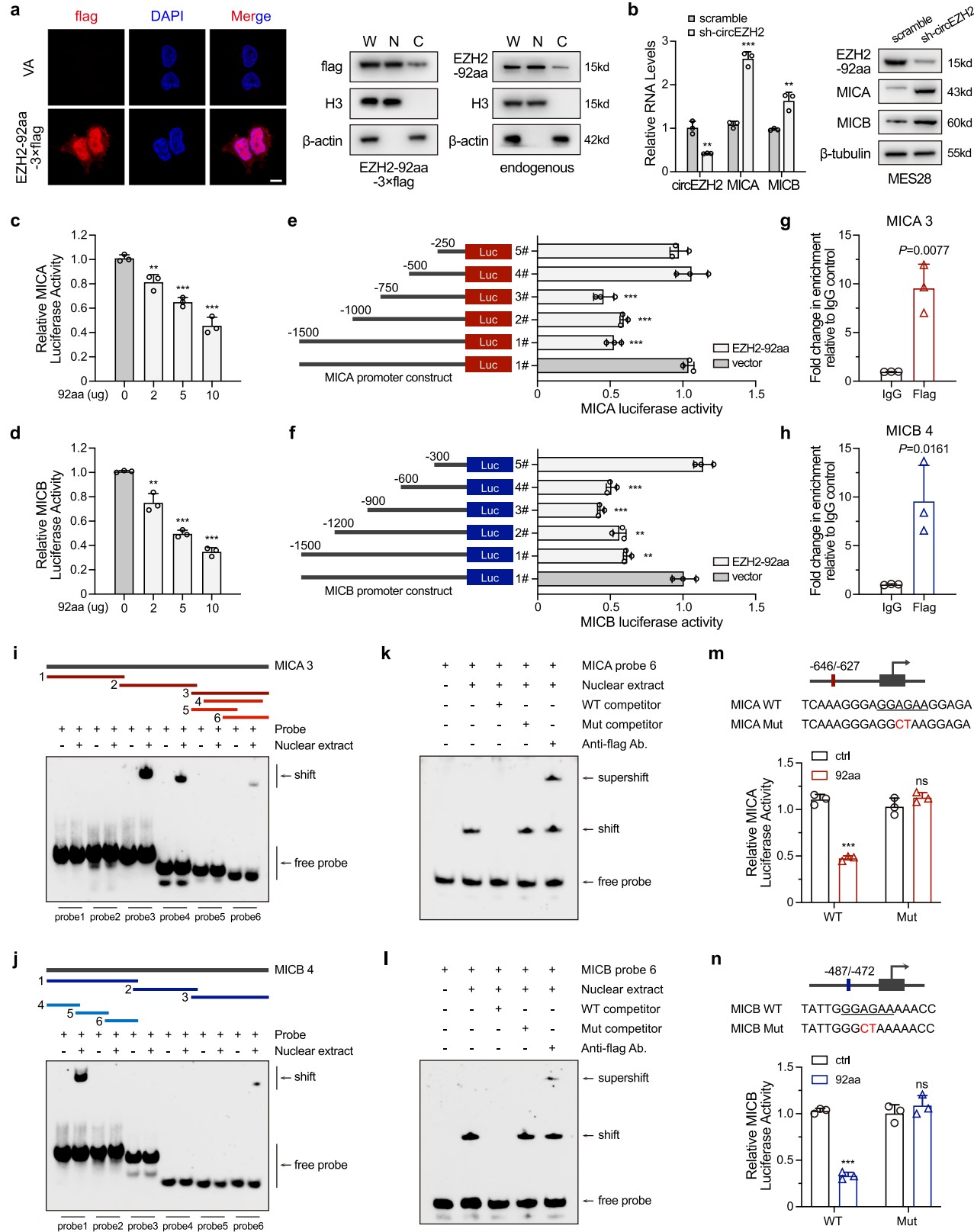

the animals (Fig. 7b). Notably, tumour growth was accelerated after 28–33 days, indicating that circEZH2 KD alone only exerted a short-term effect on suppressing brain tumour progression in the mice (Fig. 7c). Interestingly, late-passage GSCs also resumed growth in vitro (Supplementary Fig. 6a, b). However, effective knockdown was confirmed in those late-passage cell lines or late-stage xenografts (Supplementary Fig. 6c, d). This difference may be attributed to the

adaptive mechanism of EZH2 degradation-induced H3K27me3 reduction, as has been reported in breast cancer[36]. Treatment of GSC-bearing mice with NK cells ($2.0 \times 10^6$ cells every 7 days) also temporally inhibited tumour progression, although tumour growth accelerated after 3 weeks (Fig. 7c). NK cell treatment prolonged OS in mice by only ~6–8 days, indicating that the resistance of GSCs to NK cells reduced cytotoxicity. The combination of circEZH2 KD and NK cell treatment,

**Fig. 5 | EZH2-92aa directly represses the transcription of the NKG2D ligands MICA/B. a** Left, representative IF image of MES28 GSCs transfected with EZH2-92aa-3×flag and stained with an anti-flag antibody. VA, vector alone. Scale bar, 5 μm. Right, immunoblot of EZH2-92aa-3×Flag or endogenous EZH2-92aa in the indicated cellular fraction of MES28 GSCs transfected with EZH2-92aa-3×Flag or GSC23 GSCs using an anti-Flag antibody or the custom anti-EZH2-92aa antibody. W, whole cell lysate; N, nuclear fraction; C, cytoplasmic fraction. **b** Left, mRNA levels of MICA/MICB in circEZH2 stable KD MES28 GSCs. Right, immunoblot showing the levels of MICA/MICB in circEZH2 stable KD MES28 GSCs. Scramble vs sh-circEZH2, circEZH2 $P = 0.0023$, MICA $P = 0.0001$, MICB $P = 0.0046$. **c, d** Luciferase activity driven by MICA/MICB in MES28 GSCs transfected with increasing doses of the EZH2-92aa OV plasmid. MICA, 0 vs 2 $P = 0.0078$, 0 vs 5 $P = 0.0002$, 0 vs 10 $P = 0.0002$; MICB, 0 vs 2 $P = 0.0042$, 0 vs 5 $P = 8.14e{-}06$, 0 vs 10 $P = 5.25e{-}06$. **e, f** Luciferase activity of the MICA/MICB promoter fragments fused to a luciferase reporter gene. MICA, vector 1# vs EZH2-92aa 1# $P = 1.49e{-}04$, vector 1# vs EZH2-92aa 2# $P = 6.92e{-}05$, vector 1# vs EZH2-92aa 3# $P = 2.11e{-}04$; MICB, vector 1# vs EZH2-92aa 1# $P = 0.0014$, vector 1# vs EZH2-92aa 2# $P = 0.0012$, vector 1# vs EZH2-92aa 3# $P = 0.0003$, vector 1# vs EZH2-92aa 4# $P = 0.0006$. **g, h** ChIP-qPCR analysis of the binding site of EZH2-92aa-3×Flag in the MICA/MICB promoters in MES28 GSCs transfected with EZH2-92aa-3×Flag. IgG vs Flag, MICA $P = 0.0077$, MICB $P = 0.0161$. **i, j** EMSA was performed using the nuclear extract of MES28 GSCs transfected with EZH2-92aa-3×Flag and 6 specific biotin-labelled MICA/MICB probes. Independent experiments were performed three times with similar results. **k, l** EMSA was performed using the nuclear extract of MES28 GSCs transfected with EZH2-92aa-3×Flag, biotin-labelled MICA/MICB probes, a 200-fold excess of unlabelled MICA/MICB probes (200× WT competitor), biotin-labelled mutated MICA/MICB probes (200× Mut competitor) and an anti-flag antibody. Independent experiments were performed three times with similar results. **m, n** Mutants of the EZH2-92aa binding site in the MICA/MICB promoter were constructed. Luciferase activity of the WT or mutated MICA/MICB constructs with EZH2-92aa-3×Flag OV. Ctrl vs 92aa, MICA WT, $P = 5.52e{-}05$; MICB WT, $P = 1.40e{-}05$. The data are pooled from three independent experiments. The data are presented as the mean ± SD. Unpaired two-tailed Student's $t$ test was used to determine the significance of differences between the indicated groups where applicable. ns, nonsignificant, *$P < 0.05$; **$P < 0.01$; ***$P < 0.001$. Source data are provided as a Source data file.

in sharp contrast, maximally inhibited intracranial GSC growth and prolonged the OS of mice to ~70–80 days (Fig. 7b, c). These data indicated that EZH2-92aa inhibition prevented NK cell resistance in GSCs and enhanced the efficiency of NK cell therapy in a mouse GBM model. After isolating tumour-infiltrating NK cells in this mouse GBM model, we characterized the markers associated with NK cell degranulation, cytokine production and activation (on day 26). The expression of CD107a, granzyme B, perforin, IFN-γ, and TNF-α was upregulated in the combination treatment group compared with the NK cell group, further indicating the sensitization of these EZH2-92aa KD GSCs to NK cell-mediated cytotoxicity (Fig. 7d, e).

Despite the prolongation of survival in the combination treatment group, none of the animals were cured, probably because some adaptive mechanisms against NK cells were activated in the residual tumour mass. We next compared the expression of the aforementioned dysfunction markers[11] in GSCs from the xenografts between the early and late stages (days 15 and 35, respectively). Among the examined markers, PD-L1 was upregulated on GSCs in late-stage tumours under NK cell pressure (Supplementary Fig. 7a, b). We also investigated the alternative factors generated from the tumour microenvironment (TME) in immunocompetent mice. C57BL/6J mice were intracranially implanted with the mouse glioma cell line GL261 with circEZH2 stable KD or control and received incremental orthotopic primary NK cell injection to establish a model comparable to the NDG model (Fig. 7f). Similar to the NDG model, the combination of EZH2-92aa inhibition and NK cell treatment substantially reduced the tumour volume and prolonged survival (Fig. 7g, h); however, the prolongation was not as apparent as in the NDG model, thus suggesting the presence of additional immunosuppressive factors from the TME. We then profiled the immune infiltrate, including subsets of myeloid and lymphoid cells, by flow cytometry (Supplementary Fig. 7c, d). Notably, M2-like tumour-associated macrophages (TAMs), including CD206- and PD-L1-positive macrophages and microglia, infiltrated the tumour mass in each group at marked frequencies in the tumour mass (day 28), regardless of EZH2-92aa KD or the administration of NK cell therapy (Supplementary Fig. 7e). Although NK cell degranulation and cytokine production were improved upon EZH2-92aa knockdown (Fig. 7i), a substantial proportion of exhausted CD8+ T cells expressing PD-1 or TIM-3 were identified (Supplementary Fig. 7f). Based on the findings described above, we speculated that the addition of an immune checkpoint blocker (ICB) might further optimize the therapeutic effect on this immunocompetent model (Fig. 7f). As expected, the addition of ICB treatment resulted in a more significantly reduced tumour volume and prolonged survival (Fig. 7g, h). Surprisingly, the combination of EZH2-92aa inhibition, NK cell injection, and the anti-PD1 antibody fully maximized survival and even cleared the tumour masses in two mice from the treated group, showing a better effect than the combined treatment with EZH2-92aa inhibition plus NK cells (Fig. 7g). The activation of NK cells and cytotoxic T cells was also sustained at a higher level in the triple combination group (Fig. 7i). Collectively, these data demonstrated that EZH2-92aa inhibition sensitized GBM to NK cell therapy and could be further boosted by immune checkpoint blockade.

## Discussion

GSCs present the greatest therapeutic challenge in GBM due to their high heterogeneity, resistance to chemotherapy/radiotherapy, and tumour reinitiation ability[37,38]. Although several reports have shown that these cancer stem-like cells are also susceptible to NK cell-mediated immune attack[4,5,17,39,40], NK cell therapy as either monotherapy or combination therapy has rarely shown satisfactory effects in GBM patients in clinical trials[10], suggesting an unknown mechanism of GSC resistance to NK cells.

The NKG2D pathway plays a critical role in cancer cell surveillance and eradication[41]. Cancer cells have also developed several mechanisms to evade NKG2D-mediated lysis by NK cells. For example, cancer cells secrete a large amount of TGF-β to block NKG2D transcription and create an immunosuppressive tumour microenvironment[13,42]. In addition, cancer cells can release NKG2D ligand-containing exosomes to evade detection by NK cells[43,44]. Metabolic dysregulation, such as LDH overexpression, can induce NKG2D ligand expression on myeloid cells and subvert NK cell antitumor responses[45]. Our finding that EZH2-92aa suppressed NKG2D ligand expression on GSCs suggested a mechanism by which cancer cells evade NK cell toxicity. EZH2-92aa acts as a transcriptional repressor and directly inhibits MICA/B expression, thus inducing GSC resistance to NK cell cytotoxicity (Fig. 8). A recent report showed that a circRNA-encoded protein, Nlgn173, binds to ING4 and C8orf44-SGK3 to promote aberrant collagen deposition[46]. We inferred that given their nuclear localization, these circRNA-encoded proteins are the 'hidden transcription factors' that critically contribute to the global transcriptional network. Furthermore, EZH2-92aa was overexpressed in clinical GBM tumours and predicted worse OS of GBM patients, suggesting the prognostic role of this protein. Despite these findings, strategies to target EZH2-92aa in GSCs, such as small molecule inhibitors, require further investigation.

Although circRNAs were previously considered noncoding RNAs, we and several groups systematically revealed that circRNAs can function as protein templates[14,20]. To date, we have identified two major mechanisms by which these circRNA-encoded proteins perform their biological functions. First, these proteins have functions distinct

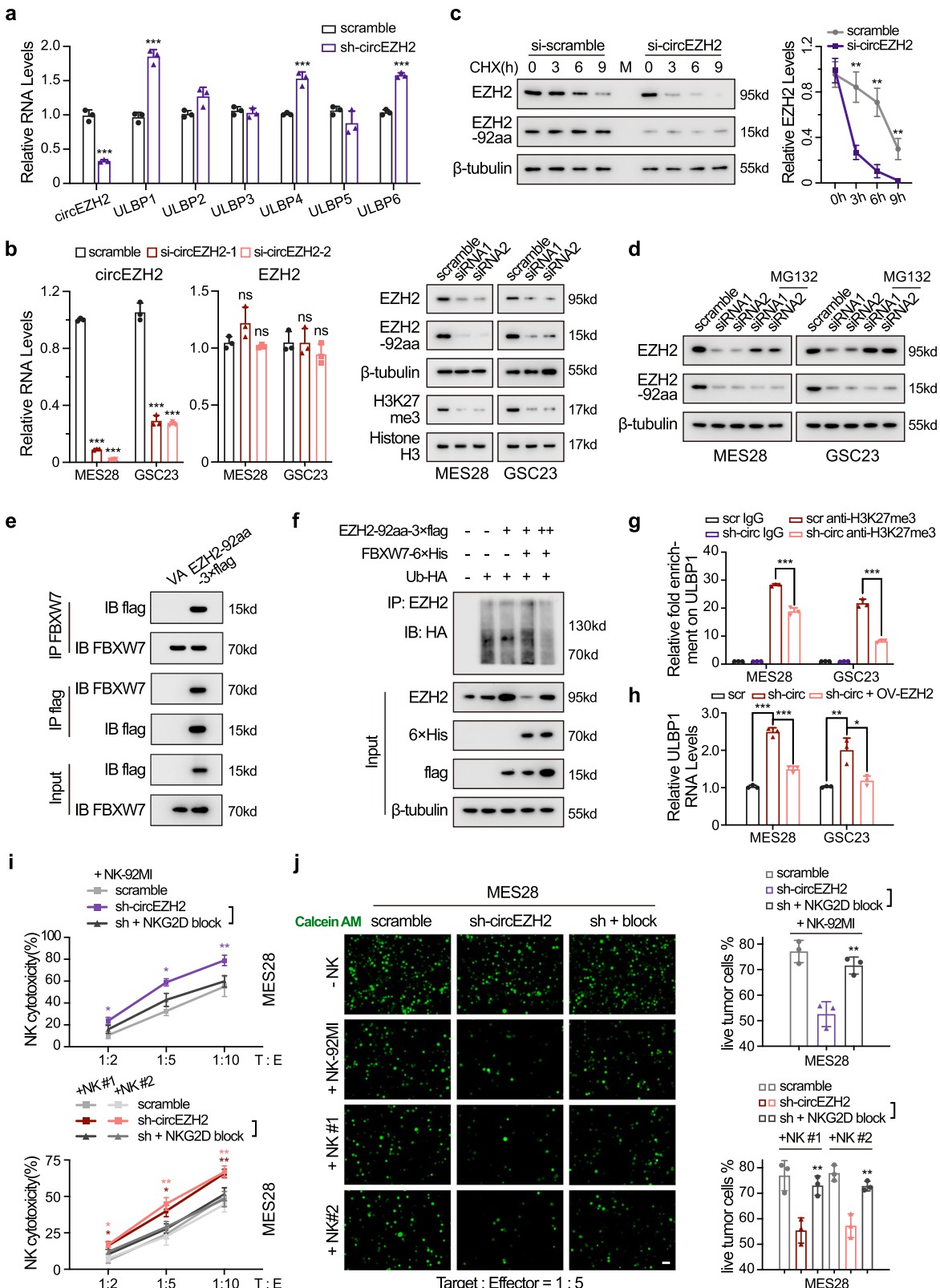

from those of their cognate genes; for example, C-E-cad enhances EGFR-STAT3 signalling in GBM to promote GSC stemness[14]. Similarly, the circARHGAP35 protein promotes cancer cell progression by interacting with TFII-I in the nucleus[20]. Second, circRNA-encoded proteins can perform functions similar to those of the corresponding full-length proteins, primarily by protecting the full-length proteins from degradation[33,34]. In this study, we identified that in addition to

serving as a transcriptional repressor, EZH2-92aa also protects the EZH2 protein from FBXW7-induced degradation (Fig. 8). Moreover, EZH2 is also an oncogene in many cancers, including GBM[47,48]. Via DNA methylation or modifications of histone proteins, EZH2 influences several aspects of cellular physiology and function. Indeed, in EZH2-92aa KD GSCs, we observed a short-term growth reduction in vitro, but this effect was sustained for at most 3-4 weeks (Supplementary

**Fig. 6 | EZH2-92aa indirectly represses the NKG2D ligand ULBP1 by stabilizing EZH2. a** Relative RNA levels of ULBPs (ULBP1-6) in circEZH2 stable KD MES28 GSCs. Scramble vs sh-circEZH2, circEZH2 $P = 1.68e-04$, ULBP1 $P = 2.49e-04$, ULBP4 $P = 8.36e-04$, ULBP6 $P = 6.42e-05$. **b** Relative RNA and protein levels of circEZH2/ EZH2-92aa and EZH2 were measured in MES28 and GSC23 GSCs transfected with circEZH2-siRNAs or control scrambled siRNAs. H3K27me3 protein levels were also determined. MES28, scramble vs si-circEZH2-1, $P = 1.87e-08$, si-circEZH2-2, $P = 9.90e-09$; GSC23, scramble vs si-circEZH2-1, $P = 6.00e-05$, si-circEZH2-2, $P = 3.25e-05$. **c** Left, half-life of EZH2 in circEZH2-siRNA-transfected MES28 GSCs. Right, quantitative analysis of the immunoblotting data by greyscale analysis at the indicated time points. CHX = cycloheximide. Scramble vs sh-circEZH2, 3 h $P = 0.0025$, 6 h $P = 0.0015$, 9 h $P = 0.0069$. **d** Protein level of EZH2 in circEZH2-siRNA-transfected MES28 and GSC23 GSCs after MG132 treatment (20 μM) for 6 h. **e** 293T cells were transfected with EZH2-92aa-3xflag, and total cell lysates were subjected to immunoprecipitation using anti-Flag and anti-FBXW7 antibodies, followed by immunoblotting with an anti-Flag or anti-FBXW7 antibody. **f** EZH2-92aa-3xFlag and FBXW7-6xHis were transfected into 293T cells as indicated in combination with Ub-HA. After treatment with MG132 (20 μM) for 6 h, total cell lysates were subjected to immunoprecipitation using an anti-EZH2 antibody, followed by immunoblotting with an anti-HA antibody. **g** MES28 and GSC23 circEZH2 stable KD

or control cells were analysed for H3K27me3 levels in the ULBP1 promoter using a ChIP assay. Relative fold changes compared with IgG are shown. Scramble anti-H3K27me3 vs sh-circEZH2 anti-H3K27me3, MES28 $P = 3.07e-04$, GSC23 $1.15e-04$. **h** Relative RNA levels of ULBP1 in MES28 and GSC23 GSCs transfected with scrambled shRNA control, sh-circEZH2 or sh-circEZH2 together with the EZH2 OV plasmid. Scramble vs sh-circEZH2, MES28 $P = 3.25e-05$, GSC23 $P = 0.0067$; sh-circEZH2 vs sh-circEZH2+OV-EZH2, MES29 $P = 2.65e-04$, GSC23 $P = 0.0157$. **i** Cytotoxicity of NK cells after coculture with MES28 GSCs with stable circEZH2 KD, stable circEZH2 KD plus NKG2D block (10 μg/ml). Sh-circEZH2 vs sh-circEZH2 plus block, NK-92MI, 1:2 $P = 0.0481$, 1:5 $P = 0.0147$, 1:10 $P = 0.0092$; NK #1, 1:2 $P = 0.0238$, 1:5 $P = 0.0155$, 1:10 $P = 0.0097$; NK #2, 1:2 $P = 0.0231$, 1:5 $P = 0.0086$, 1:10 $P = 0.0087$. **j** Left panel, images showing remaining live MES28 GSCs with indicated modifications after coculture with NK cells using calcein AM staining. NKG2D block (10 μg/ml) was added in the indicated groups. Scale bar, 50 μm. Right panel, quantification of the remaining live GSCs. Sh-circEZH2 vs sh-circEZH2 plus block, NK-92MI, $P = 0.0051$; NK #1, $P = 0.0081$; NK #2, $P = 0.0059$. The data are presented as the mean ± SD from three independent experiments. Unpaired two-tailed Student's $t$ test was used to determine the significance of differences between the indicated groups where applicable. ns, nonsignificant, *$P < 0.05$; **$P < 0.01$; ***$P < 0.001$. Source data are provided as a Source data file.

Fig. 6a–c). Similarly, the in vivo tumorigenesis assay indicated that EZH2-92aa KD alone transiently inhibited brain tumour growth in mice (for ~20 days, Fig. 7c). However, tumour growth subsequently recovered almost to the level in control mice. In addition, EZH2-92aa KD in GSCs (GSC456 and GSC4121) with relatively low expression of EZH2-92aa (Figs. 1j, 2f and Supplementary Fig. 8a, b) did not manifest significant differences in proliferation and tumorigenesis but still resulted in moderately increased GSC sensitivity to NK cytotoxicity in vitro or in vivo (Supplementary Fig. 8c–g). These results indicated that the primary role of EZH2-92aa is mediating GSC immune evasion. Similar results of transient growth inhibition were obtained when mice were solely treated with NK cells. Our results suggested that an effective combination therapy including both targeted therapy and immunotherapy (NK cell therapy and ICB) may result in a better response in intracranial tumours.

To date, ORFs, m6A modifications, and IRESs in circRNA sequences are considered essential factors for the circRNA cap-independent translation mechanism. In the current study, we provided evidence that DDX3, an RNA helicase of the DEAD-box family, is involved in circRNA translation. A previous report showed that DDX3 depletion slowed ribosome movement and impaired elongation[49]. In addition, DDX3 can bind to circ-CTNNB1 and facilitate the transcriptional activity of YY1[27]. Together with our findings, these data suggest the unrevealed role of DDX3 in circRNA translation. Given the pattern of DDX3 overexpression in many cancers (including GBM), the targeting of aberrant DDX3 expression could be an alternative strategy for modulating a series of circRNA-encoded oncogenic proteins.

In summary, we reported a mechanism of immune evasion by GSCs and provided a potential optimized NK cell-directed immunotherapy for GBM intervention.

## Methods
### Mice and animal housing
Six-week-old female NOD.CB17-$Prkdc^{scid}Il2rg^{tm1}$/Bcgen (B-NDG; $Prkdc^{(-/-)}$, $IL2rg(X^-/X^-)$) mice were purchased from Jiangsu Biocytogen (Cat. 110586). Six-week-old female C57BL/6 mice were purchased from the Laboratory Animal Centre of the First Affiliated Hospital of Sun Yat-sen University. The animals were housed in a specific pathogen-free facility under a 12-h light-dark cycle. The temperature ranged from 24 to 26 °C and the humidity ranged from 50 to 70%. All of the animal experiments conducted in this study were approved by the Ethics Institutional Review Boards of the First Affiliated Hospital of Sun Yat-sen University (Approval No. [2021]171 and [2021]173).

### Human high-grade glioma, paired adjacent samples and blood samples
Human high-grade glioma tumour and adjacent tissues, as well as peripheral blood, were collected at the Department of Neurosurgery of the First Affiliated Hospital of Sun Yat-sen University after obtaining patient consent and after confirmation by neuropathologists. Patients' consent to publish clinical information potentially identifying individuals were also obtained. The study was approved by the Ethics Institutional Review Boards of the First Affiliated Hospital of Sun Yat-sen University (Approval No. [2020]322) and complied with all relevant ethical regulations regarding human participants.

### Cell lines and cell culture
GSC lines, including MES28, GSC23, 456, 387, 4121 and 3691, were kindly provided by Dr. Jeremy N. Rich (UPMC). These cell lines were cultured in neurobasal medium (Gibco) supplemented with B27 (Life Technologies), bFGF and EGF (both 20 μg/ml, R&D Systems). NHAs were purchased from Lonza and cultured with an AGM™ Astrocyte Growth Medium Bullet Kit (Lonza) according to the manufacturer's recommendation. NK-92MI cells (ATCC, CRL-2408) were cultured in NK complete medium (alpha minimum essential medium supplemented with 2 mM L-glutamine, 1.5 g/L sodium bicarbonate, 0.2 mM inositol, 0.1 mM 2-mercaptoethanol, 0.02 mM folic acid, 12.5% horse serum and 12.5% foetal bovine serum (FBS) but without ribonucleosides and deoxyribonucleosides) as previously reported[32]. 293T cells (ATCC, CRL-3216) and GL261 (DSMZ, ACC 802) were cultured in DMEM (Gibco) supplemented with 10% FBS according to standard protocols. The cell lines were authenticated using short tandem repeat (STR) fingerprinting method.

### Isolation of human primary NK cells
PBMCs were isolated from human peripheral blood by Ficoll (17144002, Cytiva) density gradient centrifugation. NK cells were sorted on a BD FACSAria II flow cytometer with a gating strategy of live $^+$CD45$^+$CD56$^+$CD3$^-$. The sorted cells were cultured in complete RPMI 1640 medium (Gibco) containing IL-2 (100 U/ml, GenScript) and IL-15 (5 ng/ml, Peprotech), as described in a previous report[50].

### Preparation of murine primary NK cells for intracranial injection
Murine spleens were collected, minced with a syringe plunger, washed with PBS, and the cell suspension was collected. Murine primary NK cells were isolated by negative depletion using a Mojosort Mouse NK Cell Isolation Kit (480049, Biolegend) and cultured in RPMI 1640

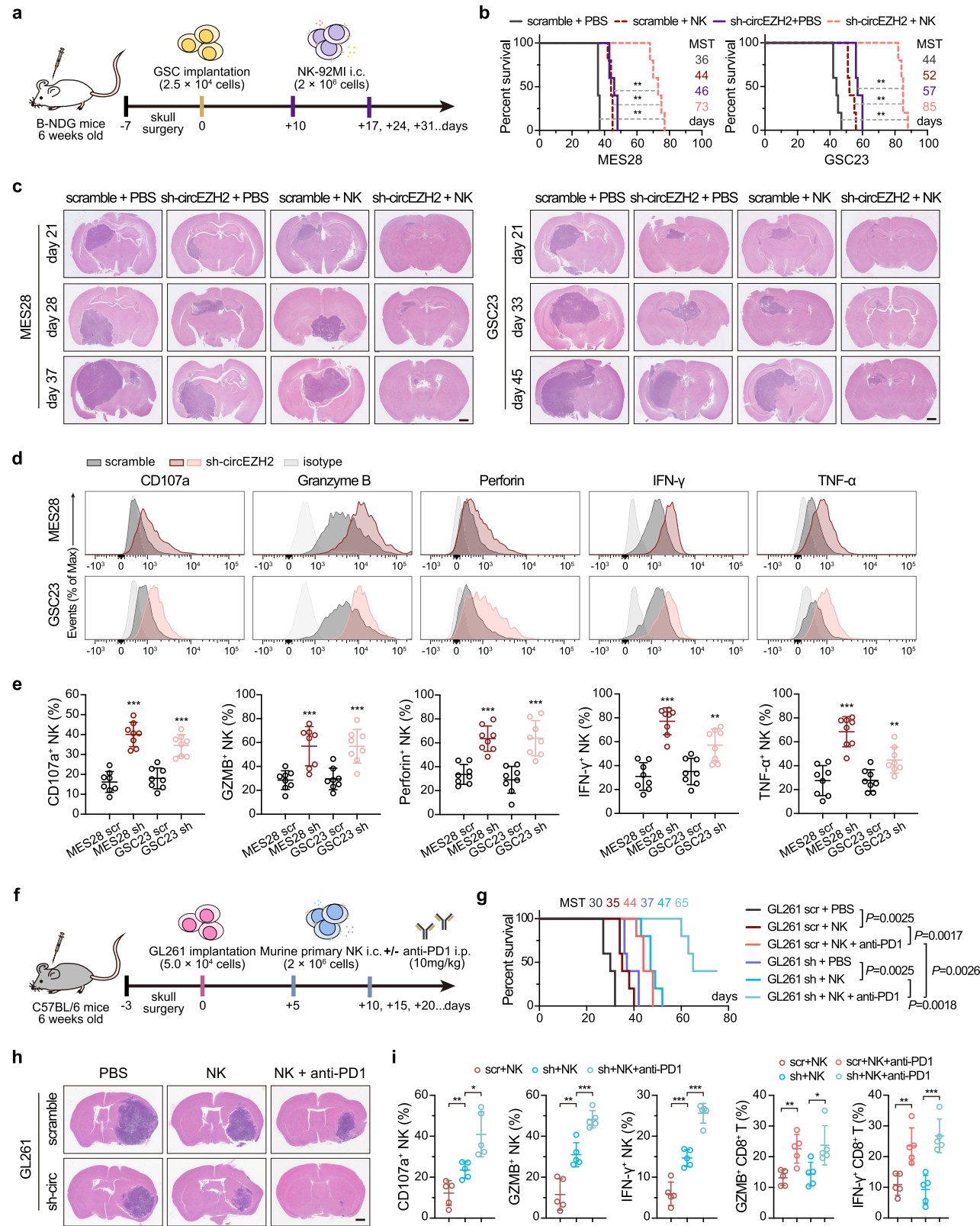

medium (Gibco) supplemented with 10% FBS and IL-15 (1 ng/ml, Peprotech) in accordance with a previously reported procedure[51]. NK cells were activated with IL-12 (25 ng/ml, R&D Systems), IL-15 (50 ng/ml) and IL-18 (5 ng/ml, R&D Systems) one night before the intracranial injection. The cells were washed and resuspended in PBS prior to injection.

**Xenograft studies**

An orthotopic xenograft model was established in this study as previously described[17,52]. Mice were randomly allocated to treatment or control groups. Animals were first anaesthetized with isoflurane. CircEZH2 stable KD or control GSCs ($2.5 \times 10^4$) or GL261 cells ($5.0 \times 10^4$) in 5 µl of PBS were then intracranially injected using a 10-µl Hamilton

**Fig. 7 | Inhibition of EZH2-92aa sensitizes GSCs to NK cell cytotoxicity and synergizes with immune checkpoint blockade. a** Illustration of the in situ GBM model in immunocompromised mice and treatment with NK cells. **b** Kaplan–Meier analysis of mice intracranially implanted with MES28 and GSC23 GSCs with stable KD of circEZH2 or control cells and treated with PBS or NK cells (*n* = 5 per group). Log-rank test. **\*\*P* < 0.01. MST, median survival time. Scale bar, 1 mm. Sh-circEZH2+NK vs scramble, MES28 *P* = 0.0023, GSC23 *P* = 0.0017; sh-circEZH2+NK vs scramble+NK, MES28 *P* = 0.0026, GSC23 *P* = 0.0017; sh-circEZH2+NK vs sh-circEZH2, MES28 *P* = 0.0025, GSC23 *P* = 0.0025. **c** Representative H&E-stained brain slices from mice with indicated treatment in (**b**). Scale bar, 1 mm. **d**, **e** Representative histogram (**d**) and quantification (**e**) of the expression of the indicated molecules in NK cells isolated from the abovementioned GBM model (*n* = 8 per group). scr vs sh, MES28, CD107a *P* = 7.37e−07, GZMB *P* = 5.92e−04, Perforin *P* = 1.97e−05, IFN-γ *P* = 1.17e−06, TNF-α *P* = 1.36e−05; GSC23, CD107a *P* = 2.21e−05, GZMB *P* = 3.96e−04, Perforin *P* = 1.09e−04, IFN-γ *P* = 0.0028, TNF-α *P* = 0.0040. **f** Illustration of the in situ GBM model in C57BL/6 mice and treatment

with NK cells and the anti-PD1 antibody. **g** Kaplan–Meier analysis of mice intracranially implanted with GL261 cells with stable circEZH2 KD or scrambled control cells and treated with PBS, NK cells or NK cells combined with an anti-PD1 antibody (*n* = 5 per group). The log-rank test was performed between the indicated groups. MST, median survival time. **h** Representative images of H&E-stained brain slices from mice with indicated treatment in (**g**). Scale bar, 1 mm. **i** Frequencies of NK or CD8⁺ T cells positive with the indicated molecules isolated from the tumour mass from the abovementioned GL261 model (*n* = 5 per group). For NK cell panel, scr+NK vs sh+NK, CD107a *P* = 0.0085, GZMB *P* = 0.0020, IFN-γ *P* = 0.0006; sh+NK vs sh +NK + anti-PD1, CD107a *P* = 0.0102, GZMB *P* = 0.0007, IFN-γ *P* = 5.61e−05; for CD8 T cell panel, scr+NK vs scr+NK + anti-PD1, GZMB *P* = 0.0041, IFN-γ *P* = 0.0035; sh+NK vs sh+NK + anti-PD1, GZMB *P* = 0.0224, IFN-γ *P* = 0.0006. The data are presented as the mean ± SD values. Unpaired two-tailed Student's *t* test was used to determine the significance of differences between the indicated groups. ns, nonsignificant, \**P* < 0.05; \*\**P* < 0.01; \*\*\**P* < 0.001. Source data are provided as a Source data file.

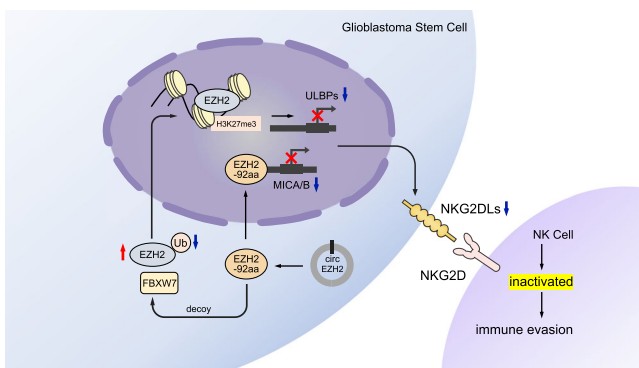

**Fig. 8 | Graphical abstract.** The diagram illustrates the mechanism underlying EZH2-92aa-mediated immune escape of GSCs from NK cells. EZH-92aa, encoded by circEZH2 in GSCs, can not only directly bind to the promoters of MICA/B and represses their transcription, but also serves as a decoy of E3-ligase FBXW7 to protect EZH2 from degradation and indirectly impedes the expression of ULBPs. The downregulation of NKG2DLs (MICA/B and ULBPs) caused by EZH2-92aa is responsible for the immune evasion of GSCs from NK cytotoxicity.

syringe through a guide screw into the right frontal lobe at a depth of 3 mm. Ten days after inoculation, 2 × 10⁶ NK-92MI cells (CRL-2408, ATCC) or primary murine NK cells in 5 µl of PBS were injected intracranially every 7 or 5 days via the same guide screw at the same depth. For the GL261 model, the mice also received an intraperitoneal injection of anti-PD1 antibody (BE0146, BioXcell, 10 mg/kg, 1:5 diluted in PBS) or PBS on the same day as the intracranial injection of NK cells. The mice were observed and weighed daily, and humanely euthanized once they presented neurological symptoms, displayed 20% weight loss or became moribund, in order to ensure that the intracranial tumour burden would not exceed the ethical limits. This was permitted by the Ethics Institutional Review Boards of the First Affiliated Hospital of Sun Yat-sen University.

## Library construction and sequencing of brain and cell line samples

Total RNA was extracted by using a TRIzol reagent kit (Invitrogen, Carlsbad, CA, USA) according to the manufacturer's protocol. RNA quality was assessed with an Agilent 2100 Bioanalyzer (Agilent Technologies, Palo Alto, CA, USA) and RNase-free agarose gel electrophoresis. Next, a strand-specific rRNA-depleted RNA-seq library was constructed by using a VAHTS Total RNA-seq (H/M/R) Library Prep Kit (Vazyme, Nanjing, China). Briefly, rRNAs were removed, and the retained RNAs were fragmented into short fragments with fragmentation buffer and reverse transcribed into cDNAs with random primers. Second-strand cDNAs were synthesized in a mixture containing DNA

polymerase I, RNase H, dNTPs (dUTP instead of dTTP) and buffer. Afterwards, the cDNA fragments were purified with VAHTS™ DNA Clean Beads, end-repaired, and the poly(A) tail was added before being ligated to Illumina sequencing adaptors. Subsequently, uracil-N-glycosylase (UNG) was used to digest the second-strand cDNAs. The digested products were purified with VAHTS™ DNA Clean Beads and amplified by PCR to complete the library construction.

A linear RNA-depleted RNA-seq library was also constructed for each sample to increase the sensitivity of detection for low-abundance circRNAs. Total RNA was treated with RNase R to degrade the linear RNAs and purified by using an RNeasy MinElute Cleanup Kit (Qiagen, Venlo, The Netherlands). The subsequent approach was the same as the procedure used to construct the strand-specific rRNA-depleted RNA-seq library mentioned above. All of the libraries were sequenced on the Illumina HiSeq X10 platform by Gene Denovo Honour (Diyao) Biotechnology Co. (Guangzhou, China). All of the data described above were deposited in the NCBI database under the accession ID PRJNA525736.

For the construction of circEZH2 stable KD and control cell lines, polyA mRNA was enriched by Oligo(dT) beads. Subsequently, the enriched mRNA was fragmented into short fragments using fragmentation buffer and reverse transcribed into cDNAs with the NEBNext Ultra RNA Library Prep Kit for Illumina (NEB #7530, New England Biolabs, Ipswich, MA, USA). The purified double-stranded cDNA fragments were end-repaired, A bases were added, and the fragments were ligated to Illumina sequencing adaptors. The ligation reaction was purified with AMPure XP Beads (1.0X). Ligated fragments were subjected to size selection via agarose gel electrophoresis and PCR amplification. The cDNA library was sequenced on the Illumina HiSeq X10 platform. These data were deposited in the NCBI database under the accession ID PRJNA862279.

## Bioinformatics analysis

Raw reads were filtered with fastp[53] (version 0.18.0) to obtain high-quality clean reads. Bowtie2[54] (version 2.3.0) was used to map reads to the ribosomal RNA (rRNA) database. The rRNA mapped reads were then removed. The remaining reads were subsequently used in the circRNA and coding gene analyses.

For linear RNA-depleted RNA-seq data, the rRNA-removed reads were then mapped to a reference genome (human genome hg38) using TopHat2 (version 2.1.1)[55]. Reads that mapped to genomes were discarded according to the mapping information recorded in bam files using in-house Perl scripts, and unmapped reads were then collected for circRNA identification. Next, 20mers from both ends of the unmapped reads were extracted using in-house Perl scripts and aligned to the reference genome (bowtie2, version 2.3.0) to locate unique anchor positions within splice sites. Anchor reads that aligned in the reverse orientation (head-to-tail)

indicated circRNA splicing and were then subjected to find_circ (version 1.2, https://github.com/marvin-jens/find_circ/) to identify circRNAs. The anchor alignments were then extended such that the completely aligned reads and the breakpoints were flanked by GU/AG splice sites. A candidate circRNA was called if it was supported by at least two unique back-spliced reads from at least one sample. CircRNAs were also blasted against the circBase database to determine if they have been reported. CircRNAs that were not annotated were defined as newly discovered circRNAs. Back-spliced junction reads were scaled to RPM (reads per million mapped reads) to quantify circRNAs. The default parameters of edgeR were used, and differentially expressed genes (DEGs) were selected based on a log2 fold-change ≥1 and $P$ value < 0.05. The circORFs in circRNAs were predicted with the method reported by Pamudurti et al.[21], and the length of the circORFs was also calculated.

The linear RNA-retained RNA-seq data were used to analyse the expression levels of the coding genes. Data were mapped to the reference genome with TopHat2 (version 2.1.1), and transcript abundances were quantified with RSEM software[56] (version 1.2.19). The transcript expression level was normalized by using the FPKM (fragments per kilobase of transcript per million mapped reads) method. The values of transcripts from the same gene were merged to obtain read counts and expression levels at the gene level. The default parameters of edgeR were used, and DEGs were selected based on a log2 fold-change ≥1 and FDR < 0.05.

The expression levels of target circRNAs were used as a molecular phenotype, and Pearson correlation coefficients between target circRNAs and each coding gene were calculated to determine the relationship between target circRNAs and gene sets involved in specific pathways. Next, coding genes were ranked by Pearson correlation coefficients, and GSEA was performed to screen significantly enriched pathways. The enriched pathways indicated that the expression of target circRNAs was positively or negatively correlated with the expression of the gene set involved in the pathways.

### Plasmids and transfection

The circEZH2 OV plasmid, circEZH2 del-ATG plasmid (mut), EZH2-92aa-3×Flag ORF plasmid, EZH2 OV plasmid, FBXW7-6×His OV plasmid and Ubiquitin-HA OV plasmid were constructed by chemical gene synthesis (Generay Biotech, Shanghai, China) employing pCDH-CMV-MCS-EF1copGFP-T2A-Puro or pcDNA3.1(+) as the backbone. Plasmids were transfected using Lipofectamine 3000 (Invitrogen) according to the manufacturer's protocol. For IRES activity validation, the EMCV-IRES sequence, putative circEZH2 IRES sequence (40-196) and IRES-Del sequences were inserted into the multiple cloning site (MCS) located between the segments of the Rluc coding sequence ('uc' and 'RL') in the Circ-Rluc-IRES-Reporter vector.

### RNA interference and transfection

SiRNAs were obtained from GenePharma (Jiangsu, China). Target sequences are listed in Supplementary Table 3. Lipofectamine RNAi-MAX Transfection Reagent (Invitrogen) was used to conduct transfection according to the manufacturer's protocols.

### Lentivirus production and stable cell line establishment

Lentiviral shRNA vectors were obtained from GenePharma (Jiangsu, China). For OV experiments, lentiviral vectors expressing circEZH2, circEZH2 del-ATG and EZH2-92aa-3×Flag ORF were cotransfected with the packaging plasmid psPAX2 (Addgene) and the envelope plasmid pMD2.G (Addgene) into 293T cells using Lipofectamine 3000 (Invitrogen) for lentivirus production. To establish stable cell lines, GSCs were transduced with the above lentiviral vectors in culture medium containing 8 μg/ml polybrene (GenePharma). After 24 h of incubation, cells were screened with 2 μg/ml puromycin for 3 days.

### RNA fluorescence in situ hybridization (FISH)

Cy3-labelled oligonucleotide probes complementary to the circEZH2 junction sequence were synthesized by GenePharma (Jiangsu, China). GSCs were seeded onto coverslips pretreated with poly-L-ornithine (Sigma-Aldrich). FISH was performed using an RNA FISH kit (GenePharma) according to the manufacturer's instructions. Images were acquired with a ZEISS LSM 880 confocal microscope with an Airyscan detector. The FISH probe sequence is provided in Supplementary Table 3.

### RNase R treatment

Total RNA extracted from cells was treated with RNase R (Lucigen) at 37 °C for 15 min. RT-qPCR was then performed to validate the resistance of circRNA to RNase R digestion.

### Actinomycin D assay

293T cells were seeded into 24-well plates ($5 \times 10^4$ cells per well) and were treated with 2 μg/ml actinomycin D (HY-17559, MedChemExpress) the next day. Cells were harvested at 0, 4, 8 and 12 h after actinomycin D treatment. RT-qPCR was used to analyse the relative RNA levels of circular and linear EZH2, and the values were normalized to those in the 0-h group.

### RNA subcellular isolation

An RNA subcellular isolation kit (Active Motif) was used to isolate the cytoplasmic and nuclear RNA fractions. In brief, cells were lysed with complete lysis buffer for 10 min on ice. After centrifugation, the supernatant was collected for cytoplasmic RNA extraction, while nuclear RNA was purified from the remaining pellet. The extracted RNA was then analysed by RT-qPCR.

### Polysome profiling

293T cells were first transfected with the circEZH2 OV plasmid. Forty-eight hours later, the cells were treated with 100 μg/ml cycloheximide (CHX) in DMSO for 5 min at 37 °C, washed with CHX-containing PBS and harvested for subsequent polysome profiling. Cells were lysed in 500 μl of polysome lysis buffer [5 mM Tris-HCl (pH 7.5), 2.5 mM MgCl₂, 1.5 mM KCl, 1× EDTA-free protease inhibitor cocktail, 0.5% Triton X-100, 2 mM dithiothreitol (DTT), 0.5% sodium deoxycholate, 100 units RNase inhibitor and 100 μg/ml CHX] on ice for 15 min and were then centrifuged at 4 °C and $16,000 \times g$ for 10 min. The supernatant was then collected and overlaid onto a 5–50% (w/v) sucrose density gradient, ultracentrifuged at 4 °C and $20,000 \times g$ for 2 h in a Beckman SW41 rotor and subsequently fractionated using a BioComp PGFip Piston Gradient Fractionator Model 152. The absorbance at 254 nm was measured using an absorbance detector connected to the fraction collector. RNA was extracted from fractions using TRIzol LS solution, and RT-qPCR was conducted to evaluate the circEZH2 and EZH2 mRNA levels in the indicated fractions.

### LDH cytotoxicity assay

This assay was performed using an LDH cytotoxicity kit (C0016, Beyotime). GSCs (target cells, $5 \times 10^4$ cells/ml) were cocultured for 2 h with NK-92MI or primary NK cells (effector cells) at T:E ratios of 1:2, 1:5 and 1:10 in 96-well plates. NKG2D blocking antibody (BE0351, BioXcell) or isotype IgG control was added at a concentration of 10 μg/ml for specific experiments. After incubation, supernatants from each well were collected for further analysis and calculation according to the manufacturer's instructions.

### Calcein AM cytotoxicity assay

This assay was performed using a calcein AM staining kit (CA1630, Solarbio) according to the manufacturer's protocols. In brief, GSCs ($1 \times 10^5$ cells/ml) were seeded into 96-well plates precoated with poly-L-ornithine. The next day, 1 μM calcein AM in culture medium was added

to the cells and was then incubated for 15 min at 37 °C. Subsequently, NK-92MI or primary NK cells were seeded into the wells at a density of $0.5-1 \times 10^6$ cells/ml to achieve a 1:5 or 1:10 ratio. NKG2D blocking antibody or isotype IgG control was added at a concentration of 10 μg/ml for specific experiments. The cells were cocultured for 2 h. For complete cell lysis, GSCs were incubated with absolute ethanol. Images were acquired with an Olympus IX83 inverted microscope.

### RNA immunoprecipitation

Cells ($1 \times 10^7$) washed in PBS were resuspended in 400 μl of ice-cold PEB buffer [20 mM Tris-HCl (pH 7.5), 100 mM KCl, 5 mM $MgCl_2$, 0.5% NP40, RNase inhibitor (EO0381, Thermo Fisher Scientific) and protease inhibitor (HY-K0010, MedChemExpress)] and incubated on ice for 10 min. The lysate was centrifuged at $10,000 \times g$ for 15 min at 4 °C. The supernatant was precleared with 10 μg of control IgG for 30 min at 4 °C on a rotator. Next, the supernatant was incubated with 50 μl of Protein A-Sepharose (PAS) beads (17-1279-02, GE Healthcare) for 30 min at 4 °C with rotation. After centrifugation at $2000 \times g$ for 2 min at 4 °C, the protein concentration in the supernatant was measured with a BCA kit (Thermo Fisher Scientific) following the manufacturer's instructions. To immunoprecipitate DDX3 complexes, 1000 μg of precleared protein lysate was incubated with 50 μl of protein A agarose beads crosslinked to a rabbit polyclonal anti-DDX3 antibody for 1 h at 4 °C. Immunoprecipitated DDX3 complexes were washed 5 times with ice-cold NT2 buffer [50 mM Tris-HCl (pH 7.5), 150 mM NaCl, 1 mM $MgCl_2$ and 0.05% NP40]. RNAs associated with DDX3 were recovered with TRIzol-chloroform and analysed by RT-qPCR. BCL2 served as the positive control, as previously reported[57]. The primers are listed in Supplementary Table 4.

### RT and real-time PCR

RNA was reverse transcribed using PrimeScript RT Master Mix (RR036, Takara). qPCR was conducted with TB Green® Premix Ex Taq™ II (Tli RNaseH Plus) (RR820, Takara) in a QuantStudio 5 system (Applied Biosystems). Information about the primers is summarized in Supplementary Table 4.

### ChIP

This assay was conducted with a ChIP Kit–One Step (ab117138, Abcam) according to the manufacturer's protocols. In brief, chromatin was first extracted and sonicated in accordance with the instructions of a chromatin extraction kit (ab117152, Abcam). Reagents, including chromatin, antibodies and ChIP buffer, were added to strip wells provided in the kit and incubated for 2 h at room temperature. The wells were washed, and precipitated DNA was released by proteinase K digestion in DNA release buffer. The collected DNA was subsequently analysed by qPCR. The primers are listed in Supplementary Table 4.

### EMSA

A nuclear protein extraction kit (P0027, Beyotime) and chemiluminescent EMSA kit (GS009, Beyotime) were used in accordance with the manufacturer's protocols. A set of biotin-labelled probes was designed to target putative binding sites in the MICA or MICB promoter. Nuclear extracts from MES28 cells overexpressing EZH2-92aa-3×Flag were incubated with the indicated probes. A competition assay was conducted with either an unlabelled probe containing the WT MICA/MICB binding site or an unlabelled probe containing the mutated binding site. To identify specific DNA-binding proteins, an anti-flag antibody (F1804, Sigma-Aldrich) was employed to visualize the supershifted band. The EMSA probes are listed in Supplementary Table 5.

### Nuclear run-on (NRO) RT-qPCR

The procedure was conducted strictly according to a reported protocol[58]. Briefly, the collected cells were incubated with NP-40 lysis buffer on ice for 5 min. The lysate was centrifuged at $300 \times g$ for 4 min

at 4 °C to pellet nuclei. The nuclei were resuspended in storage buffer and mixed with reaction buffer cocktail at 30 °C for 30 min. Nuclear RNA (NRO-RNA) was extracted using the MEGAclear transcription clean-up kit (AM1908, Thermo Fisher Scientific), and genomic DNA was removed using the TURBO DNA-free kit (AM1907, Thermo Fisher Scientific). The NRO-RNA was incubated with anti-BrdU antibody (2 μg/tube, SC-32323, clone IIB5, Santa Cruz)-coated Protein G beads for 30 min at room temperature. The beads were captured by a magnet and washed prior to the extraction of NRO-RNA using TRIzol. The extracted NRO-RNA was subjected to reverse transcription and qPCR analysis. The detailed description of the protocol for preparing the agents is provided in the previously reported protocol[58].

### Dual-luciferase reporter assay

For IRES activity validation, 293T cells were transfected with the EMCV-IRES vector (positive control), empty Circ-Rluc-IRES-Reporter vector, IRES wild-type (WT) vector, or deletion vector and incubated for 48 h. Putative IRES activity was measured by a dual-luciferase assay. For promoter binding site confirmation, mutated or nonmutated fragments spanning a range from −1500 to +100 bp with respect to the transcription start site of the MICA or MICB genomic sequence were ligated into the pGL3-Basic vector. GSCs were transfected with the corresponding plasmids and Renilla luciferase by electroporation. A dual-luciferase reporter assay system (E1910, Promega) was applied based on the manufacturer's instructions. For each sample, firefly luciferase activity was normalized to Renilla luciferase activity.

### Immunoblotting

Proteins were extracted using RIPA buffer (P0013B, Beyotime) containing a protease inhibitor and were quantified with a BCA kit (Thermo Fisher Scientific). Cell or tissue lysates containing equal amounts of protein were loaded in each well of a 10–17% SDS–PAGE gel. After electrophoresis, membrane transfer and blocking, membranes were incubated with the indicated primary antibodies and HRP-conjugated secondary antibodies (31430, 31460, Invitrogen). The bands were visualized by enhanced chemiluminescence using Clarity™ Western ECL Substrate (Bio-Rad). The following antibodies were used: anti-flag (1:5000, F1804, clone M2, Sigma-Aldrich), anti-EZH2 (1:1000, 07-689, Merck Millipore), anti-H3K27me3 (1:1,000, 9733S, clone C36B11, Cell Signaling Technology), anti-Histone H3 (1:2000, 4499S, clone D1H2, Cell Signaling Technology), anti-FBXW7 (1:1000, ab109617, Abcam), anti-DDX3 (1:1000, 11115-AP, Proteintech), anti-6xHis (1:1000, ab18184, clone HIS.H8, Abcam), anti-HA (1:1000, 35534, SAB), anti-β-tubulin (1:5000, T5201, clone TUB2.1, Sigma-Aldrich), anti-β-actin (1:5000, A1978, clone AC15, Sigma-Aldrich) and HRP-conjugated secondary antibodies, including anti-rabbit IgG (1:10,000, 5220-0336, SeraCare), anti-mouse IgG (1:10,000, 5220-0341, SeraCare). A rabbit polyclonal antibody specific for EZH2-92aa (1:500) was produced by GenScript Biotech (Jiangsu, China).

### Cell suspension preparation from mouse tissue

For isolation of cells, fresh mouse brain samples were cut into pieces and digested in DMEM supplemented with collagenase IV (1 mg/ml, Gibco), DNase I (20 U/ml, Sigma-Aldrich) and hyaluronidase (0.01%, Solarbio) for 30 min at 37 °C. After digestion, the cells were filtered through a 70-μm strainer, and Debris Removal Solution (130-109-398, Miltenyi Biotec) was applied to remove myelin according to the manufacturer's instructions. Cell pellets were then treated with RBC lysis buffer (C3702, Beyotime) and resuspended in FACS staining buffer (PBS containing 2% FBS).

### Flow cytometry

The antibodies used for flow cytometry are summarized in Supplementary Table 6. Cell suspensions were surface-labelled with fluorescent antibodies for 30 min at 4 °C. For CD107a and intracellular

staining, cells were incubated with phorbol 12-myristate 13-acetate (PMA; 50 ng/ml, MedChemExpress), ionomycin (1 μg/ml, Sigma), monensin solution (00-4505-51, eBioscience), brefeldin A solution (00-4506-51, eBioscience) and PE-CD107a for 4 h at 37 °C in an incubator, as previously reported[59]. Cells were then stained for other intracellular markers with an Intracellular Fixation & Permeabilization Buffer Set (88-8824-00, eBioscience) or a Foxp3/Transcription Factor Staining Buffer Set (00-5523-00, eBioscience). Homologous IgG was used as an isotype control antibody. Dead cells were excluded using a LIVE/DEAD Fixable Near-IR/Aqua Dead Cell Stain Kit (L34975/L34965, Invitrogen) or Fixable Viability Dye eFluor 520 (65-0867-14, eBioscience). NK-92MI cells were identified with a CD45⁺CD3⁻CD56⁺ gate. The gating strategies are summarized in supplementary figures. Flow cytometry was performed with a BD LSRFortessa X-20 or Cytek Aurora instrument, and data were analysed with FlowJo software (version 10.6.2).

## Immunofluorescence

Cultured GSCs were dissociated into single cells with Accutase (Sigma-Aldrich) and seeded on poly-L-ornithine (Sigma-Aldrich)-precoated coverslips. Twenty-four hours later, the cells were sequentially fixed with 4% paraformaldehyde for 10 min, permeabilized with 0.1% Triton X-100 for 5 min at room temperature, blocked with 1% BSA in PBS, and incubated with primary antibodies at 4 °C overnight. The next day, fluorescent secondary antibodies (Invitrogen) were added and incubated for 1 h at room temperature. Nuclei were counterstained with 4′,6-diamidino-2-phenylindole (DAPI). Images were acquired using a ZEISS LSM 880 confocal microscope with an Airyscan detector.

## Immunoprecipitation

Cells were lysed with weak RIPA buffer (P0013D, Beyotime) supplemented with protease inhibitors. The supernatant was immunoprecipitated using the indicated primary antibodies on a rotator at 4 °C overnight. Then, the lysates were incubated with 40 μl of protein A/G agarose beads (Gibco) for 2 h at room temperature. The immunoprecipitates were washed five times with ice-cold PBS containing 0.1% Tween 20 (PBST) and were then subjected to SDS–PAGE and analysed by liquid chromatography-tandem MS (LC-MS/MS) or immunoblotting.

## LC-MS/MS analysis

Total protein was extracted and separated by SDS–PAGE. The band at ~15 kDa was digested and was then analysed with a QExactive mass spectrometer (Thermo Fisher Scientific). The fragment spectra were analysed using the National Center for Biotechnology Information nonredundant protein database with the Mascot search engine (Matrix Science).

## Statistical analysis

Experimental data are presented as the mean ± standard deviation (SD) of at least three biological replicates. Unpaired two-tailed Student's $t$ test was used to determine statistical significance for parametric data. Paired two-tailed Student's $t$ test was used for comparison of parametric data between high-grade glioma and paired adjacent samples. The log-rank test was applied to determine the significance of differences in survival data. A $P$ value <0.05 was considered statistically significant. The degree of significance between groups is represented as follows: $*P < 0.05$; $**P < 0.01$; and $***P < 0.001$. Statistical tests were carried out with GraphPad Prism (version 8).

## Reporting summary

Further information on research design is available in the Nature Research Reporting Summary linked to this article.

## Data availability

The sequencing data were deposited in the NCBI database under the accession ID PRJNA525736 and PRJNA862279. The raw clinical data of glioma patients (containing personal information including names, record numbers and contacts, etc.) are protected and are not available due to data privacy laws. However, de-identified clinical data with personal information removed are available and provided within the Source data file, covering information including the expression levels of EZH2-92aa and gene mutation status. The remaining data are available within the Article, Supplementary Information and Source data file. Source data are provided with this paper.

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

## Acknowledgements

We gratefully thank all patients who participated in these studies. This work was supported by the National Natural Science Distinguished Youth Foundation of China (82125024 to N.Z.), the National Natural Science Foundation of China (82072779 to X.Wang), the Science and Technology Planning Project of Guangzhou (202103000019 to N.Z.), the Academician-guided scientific and technological innovation project in Chongqing (cstc2020yszx-jcyjX0002 to Y.S.) and the Natural Science Foundation of Guangdong Province (2021A1515012223 to X.Y. and 2018A030313549 to K.H.).

## Author contributions

N.Z., Y.S. and X. Wang designed the experiments. J.Z., X.Y. and J.C. performed the experiments. J.Z., K.H., X.G., X. Wu, M.Z., H.Z., L.A., X. Wang, Y.S. and N.Z. analysed the data. J.Z., Y.S. and N.Z. wrote the manuscript. X. Wang, M.Z., H.Z. and F.X. provided scientific input and helped edit the manuscript.

## Competing interests

The authors declare no competing interests.
