## [Peer Review File · Nature Communications]

Circular EZH2-Encoded EZH2-92aa Mediates Immune Evasion in Glioblastoma via Inhibition of Surface NKG2D LigandsREVIEWER COMMENTS

Reviewer #1 (Remarks to the Author): with expertise in glioblastoma

The investigators made the interesting observations regarding the activity of NK cells in glioblastoma and using gene set enrichment analysis found that expression of circEZH2 was negatively correlated with NK cell mediated cytotoxicity in the expression analysis. This observation led to the series of experiments looking at the mechanisms of EZH2 overexpression and inhibition of NK cell activity mainly using 2 patient derived glioma stem cell lines. Additionally, TCGA data looking at the EZH2-92aa product found a negative correlation of expression with prognosis, further supporting the importance of this pathway. Additional studies showed that EZH2 inhibited the expression of the NK group 2D ligands, important for NK cell function. In vivo studies using intracranial implants of the 2 GSC lines demonstrated that NK cells + knockdown of EZH2 expression significantly increased survival although there were no cures.

This is very interesting work and does provide important information about one important component of the potential immune activity against glioblastoma, where immunotherapy efforts to date have been not very effective. However, there are some concerns about the studies performed and the conclusions that this approach should be translated for use as a therapy for patients.

1. Both GSC models used have high expression of EZH2-92aa and in vivo experiments showed that knockdown of expression led to a transient reduction in tumor cell growth in the absence of any NK cells. This does raise a question about the alteration in tumor cell function in the in vivo experiments. In this context, additional GSC models, particularly those where EZH2 expression is not elevated would be very important as a comparator where knockdown would not be expected to have much impact on survival as alternative mechanisms are likely.
2. Although a highly significant prolongation of survival was seen with the knockdown and NK cells, none of the animals was cured of the tumor. This suggests that some alternative mechanisms have been implemented so examination of these tumors, both tumor cells and microenvironment would be important.
3. The use of human GSCs is important because, as mentioned, these are the most resistant component of the human cancer to treatment, but the mouse model requires use of an immunocompromised host. Therefore, important interactions with other cellular components of the immune system (ie T cells) are lacking. The investigators may want to consider syngeneic models where comparable studies can be done in immunocompetent animals.
4. Some minor points to consider:
 - a. Page 3, line 91: "Disruption of the ligand interactions..." This is a major statement and premise for this work, but is not referenced.
 - b. Page 3, line 103: "In the current study, we found that circular EZH2 (circEZH2) expression is highly related to NK cell cytotoxicity." This is also an important statement, but it is not clear if there is a correlation or a cause and effect relationship. If known, this should be clarified.
 - c. Page 5, line 143: "the expression of identified NK cell markers, including EGRs, IFNG, TNF, and 1 GZMB (Barrow et al., 2018, Supplemental Table 4), was negatively correlated with circEZH2 expression (P<0.01)". Since a series of NK cell markers were evaluated, was a multiple comparisons correction used?
 - d. Page 5, line 169: "The above findings indicated that circEZH2 is a potential oncogenic circRNA in GBM." Oncogenic implies that this gene is responsible for causing the cancer, rather than helping malignant cells avoid immune recognition. An alternative phrase should be considered.
 - e. Page 7, line 208: "Higher EZH2-92aa expression was negatively correlated with OS, indicating that EZH2-92aa may be an independent prognostic marker in GBM". I would suggest adding a qualifier that a multivariable analysis would need to be done incorporating established prognostic factors to confirm that this is an independent prognostic factor. For example, if EZH2-92aa overexpression occurs exclusively in grade 4 IDH wildtype cancers and the data include IDH mutated grade 4 tumors, then that may be sufficient to account for the survival difference.
 - f. The final section talks about DDX2 and control of EZH2 expression. Although interesting, it was not

clear how this relates to the extensive discussion of the mechanisms of NK cell inhibition with stabilization of EZH2, ligand suppression etc. This section may be better placed earlier after the discussion of EZH2-92aa and EZH2 expression control and before the efficacy studies.

Reviewer #2 (Remarks to the Author): with expertise in glioblastoma, NK cells

The study identified a new circular EZH2 RNA (circEZH2) to be expressed at high levels in certain glioblastoma stem cells (GSCs) and confirmed circEZH2 translation and expression of the respective novel EZH2-92aa protein. shRNA-mediated knockdown of EZH2-92aa expression enhanced sensitivity of GSCs to the NK cell line NK-92MI, which was correlated to a direct repressive effect of EZH2-92aa on MICA and MICB transcription and an indirect effect on the expression of other NKG2D ligands. While it is intriguing to assume reduced MICA/B expression and an ensuing lack of NKG2D activation in NK cells as the underlying mechanism for EZH2-92aa-induced escape of GBM cells from NK cell cytotoxicity, this appears to be an over-interpretation of the actual data which lack formal proof for the importance of the MICA/B-NKG2D interaction and respective confirmation with primary human NK cells.

Major:

sh-circEZH2 mediated knockdown of EZH2-92aa expression reduced MICA/B mRNA levels in MES28 cells (Figure 4B). Data confirming that this is also the case for surface expression of MICA/B proteins are missing.

Direct proof for a major role of the MICA/B-NKG2D axis for enhanced NK cell cytotoxicity against knockdown cells is lacking. Cytotoxicity experiments with wildtype and EZH2-92aa knockdown GSCs (increased MICA/B mRNA) and NK cells in the presence of a blocking NKG2D antibody or NKG2D-Fc should be included to clarify to what extent activating receptors other than NKG2D are possibly involved, and whether enhanced NK sensitivity in the knockdown cells is indeed mainly due to restored MICA/B-NKG2D interactions.

In the absence of data on surface MICA/B protein expression and functional experiments clearly demonstrating reduced MICA/B-NKG2D interaction as the cause for NK resistance, the title of the manuscript, parts of abstract and discussion, as well as the proposed model in Figure 7F are overstating what is actually shown in the study.

Alternative triggers of NK cell activity are downregulation of MHC class I and NCR activation, which may also be affected by circEZH2/EZH2-92aa. Indeed, EZH2-92aa knockdown in MES28 cells appears to have resulted in increased NCR3LG1 (=B7H6) mRNA expression (Figure 3A), which is a ligand for NKp30. The authors should comment on possible direct or indirect effects of EZH2-92aa on NCR ligand expression. Did EZH2-92aa knockdown have any effect on MHC class I expression?

NK cell cytotoxicity in vitro and in vivo was investigated using the NK-92MI cell line as a model. NK-92 cells are known for their lack of most of the inhibitory NK cell receptors and their high intrinsic cytotoxicity, which distinguishes them from primary NK cells expanded from PBMCs. In vitro cytotoxicity experiments with wildtype and EZH2-92aa knockdown GSCs should also be performed with ex vivo expanded primary NK cells from different donors to investigate whether EZH2-92aa-mediated NK resistance is indeed a general effect or limited to the particular NK cell line used.

The data on the interaction of ectopically expressed EZH2-92aa-3xflag or the overexpressed EZH2-92aa open reading frame with MICA and MICB promoter regions and their inhibitory effects on the expression of MICA/B luciferase reporter constructs are convincing, but need to be confirmed by respective experiments demonstrating similar effects for endogenous EZH2-92aa. The EMSA experiments shown in Figure 4I and 4J should be repeated with nuclear extracts of wildtype MES28 and GSC23 cells which already have high endogenous levels of EZH2-92aa protein as shown in Figure 2E. Rabbit anti-EZH2-92aa antibody may be used for supershift experiments. Alternatively,

ChIP experiments could be performed with endogenous EZH2-92aa and the EZH2-92aa antibody (Figure 4G and 4H).

In a previous study cited by the authors expression of FBXW7-185aa, the product of a circular FBXW7 RNA, was found to inhibit proliferation of GBM cells (Yang et al. 2018). Was reduced circ-FBXW7 found in GSCs with upregulated EZH2-92aa? The authors may also want to comment on how circ-FBXW7 would fit into their hypothesis that EZH2-92aa protects EZH2 protein from FBXW7-induced degradation.

EZH2-92aa knockdown on its own transiently reduced growth of brain tumor xenografts in mice, attributed by the authors to an adaptive mechanism of EZH2 degradation-induced H3K27me3 reduction. A more simple explanation would be that the growth disadvantage of EZH2-92aa knockdown cells favors the selective outgrowth of a subpopulation with less complete knockdown or loss of the shRNA construct. Accordingly, an experiment is missing comparing EZH2-92aa expression in knockdown cells from tissue culture and tumor cells growing out in the respective in vivo experiment. Did the authors confirm continued EZH2-92aa knockdown in late passage GSCs that resumed normal in vitro growth characteristics (Figure S5)?

In the last part of the study the authors show an RNA immunoprecipitation experiment indicating direct interaction of DDX3 with circEZH2 RNA and demonstrate reduced EZH2-92aa protein levels upon DDX3 knockdown, concluding from this a direct involvement of DDX3 in the regulation of circEZH2 RNA translation. This should be worded more carefully since DDX3 is involved in many different cellular processes, and its knockdown may also have global effects that indirectly inhibit EZH2-92aa protein expression. Along those lines, it appears that at least in GSC23 cells DDX3 knockdown also reduced expression of beta-actin (Figure 7C). An experiment demonstrating enhanced MICA/B expression and NK sensitivity upon DDX3 knockdown as expected with the ensuing reduction in EZH2-92aa would connect this part of the study better to the rest of it.

Minor:

References should be checked for accuracy. Some references indicated with a single author are incomplete (see for example Eisele 2006).

In text citations should be adjusted to journal style. Some appear to refer to a certain page number.

Line 25: Typing error: "vin".

Lines 347 to 348: "...resistance to GSCs 'induced' cellular toxicity". Was 'reduced' meant?

Reviewer #3 (Remarks to the Author): with expertise in circRNA (bioinformatics)

In this manuscript, the authors characterized EZH2-92aa, a novel protein encoded 44 by circular EZH2 (circEZH2), in GBM, and demonstrated that EZH2-92aa induces the 45 resistance of glioblastoma stem cells (GSCs) to NK cells. They further showed that stable EZH2-92aa knockdown enhanced NK cell-mediated eradication of 52 GSCs both in vitro and in vivo. The manuscript is overall well-organized, and the experimental data is solid and convincing. The reviewer has several comments for the authors to further improve the manuscript:

1. Detailed information for Bioinformatics analysis is needed. For example, what is the accession number for the RNA-seq data generated in this manuscript – it seems that the link provided [<https://www.ncbi.nlm.nih.gov/sra/?term=SRP095744>] included only two samples, and it is related to FBXW7 circRNA, not the current ones. How did they quantify circRNAs (with which algorithm)? How do they identify differentially expressed circRNAs (Figure 1A)? How did they quantify the expression

of circEZH2 in the clinical patient cohort (Figure 1J)? Several other bioinformatics analyses are lacking the details which need to be specified.

2. How do they characterize the length of circRNAs (Figure 1B)? How to interpret the association between circEZH2 and NK markers (Figure 1F), since many markers are actually negatively correlated with circEZH2. Furthermore, did the authors check the associations between circEZH2 with other immune cells, for example, T cells. Is it possible for circEZH2 to exert functions through T cells?

3. The authors showed the consistency between circEZH2 with circBase data – which is actually not a cancer circRNA database. Did the authors compare with other data resources, such as MiOncoCirc, CircRic, CSCD/CSCD2?

4. It is interesting to see the inhibition of EZH2-92aa sensitizes GSCs to NK cell cytotoxicity. However, the mechanism for this section need further efforts to make it clear.

Reviewer #4 (Remarks to the Author): with expertise in circRNA

This study identified a circular RNA (circEZH2) that was upregulated in Glioblastoma (GBM). circEZH2 was found to encode a polypeptide EZH2-92aa. The authors provided a series of data to support the coding ability of circEZH2 and the real presence of EZH2-92aa. Further characterizations of EZH2-92aa demonstrated that EZH2-92aa mediated NK cell tolerance of GBM through transcriptional regulation of MICA/B expression and posttranscriptional regulation of EZH protein degradation. Analyses of human samples and public data along with mice models were also used to demonstrate effects of EZH2-92aa in GBM immune evasion.

Generally this study is very well designed and the data are compelling to make the conclusions. I only have three minor concerns.

1, For the part of EZH2-92aa suppresses the transcription of MICA/B, it is better to perform some nuclear run-on assays, under EZH2-92aa overexpression and knockdown conditions, respectively.

2, Please provide more details about circEZH2. Information about: conservancy (mice circEZH2, existence or coding potency?), the possible presence of Exon-intron circRNA (ElciRNA) isoform of circEZH2 (and if there is ElciEZH2, please at least discuss its potential role in transcriptional regulation), how about the ENNHGPDWEEI epitope (is it unique in sequences in human?), how about functional domain or motif in EZH2-92aa? How about the copy numbers per cell of circEZH2?

3, The authors made a strong argument about roles of EZH2-92aa, although it would not be appropriate to just assume that circEZH2 only serves as the translational template for EZH2-92aa, without any other role as a circRNA (or even the generation and metabolism of this RNA may play some role in GBM). Please discuss this kind of possibility and tune down the significance of EZH2-92aa (such as “provided a potentially important combination strategy for GBM intervention”).

**Point-by-point responses to the reviewers' comments for NCOMMS-21-42988A**

**Manuscript NCOMMS-21-42988**

**REVIEWER COMMENTS**

**Reviewer #1 (Remarks to the Author): with expertise in glioblastoma**

The investigators made the interesting observations regarding the activity of NK cells
in glioblastoma and using gene set enrichment analysis found that expression of
circEZH2 was negatively correlated with NK cell mediated cytotoxicity in the
expression analysis. This observation led to the series of experiments looking at the
mechanisms of EZH2 overexpression and inhibition of NK cell activity mainly using 2
patient derived glioma stem cell lines. Additionally, TCGA data looking at the EZH2-
92aa product found a negative correlation of expression with prognosis, further
supporting the importance of this pathway. Additional studies showed that EZH2
inhibited the expression of the NK group 2D ligands, important for NK cell function. In
in vivo studies using intracranial implants of the 2 GSC lines demonstrated that NK cells
+ knockdown of EZH2 expression significantly increased survival although there were
no cures.

This is very interesting work and does provide important information about one
important component of the potential immune activity against glioblastoma, where
immunotherapy efforts to date have been not very effective. However, there are some
concerns about the studies performed and the conclusions that this approach should be
translated for use as a therapy for patients.

1. Both GSC models used have high expression of EZH2-92aa and in vivo experiments
showed that knockdown of expression led to a transient reduction in tumor cell growth
in the absence of any NK cells. This does raise a question about the alteration in tumor
cell function in the in vivo experiments. In this context, additional GSC models,

particularly those where EZH2 expression is not elevated would be very important as a
comparator where knockdown would not be expected to have much impact on survival
as alternative mechanisms are likely.

**Response:**

We sincerely thank the reviewer for this important suggestion. To address this, we
applied two other GSC cell lines (GSC456 and GSC4121) with relatively low circEZH2
and EZH2-92aa expression (Fig. 1j, 2f) and established stable circEZH2 KD cells
(Supplementary Fig. 8a, b). These stable circEZH2 KDs in these two GSCs did not
manifest significant differences in proliferation *in vitro* (Supplementary Fig. 8c, d)
compared with MES28 and GSC23 (Supplementary Figure 6a, b), suggesting that
cellular proliferation in these two GSCs does not depend on circEZH2. In addition,
circEZH2 KD did not significantly alter the tumorigenesis in NSG mice
(Supplementary Fig. 8f, g). However, when NK cells were present, stable circEZH2
KD still led to an improvement in GSC sensitivity to NK cells (Supplementary Fig. 8e-
45 g). The data described above as well as previously results together indicate that the
46 primary role of EZH2-92aa is mediating GSC immune evasion, and it could be
combined with NK cell therapy or immune checkpoint inhibitors (see the results
described below) to achieve a better response to treatment in patients with GBM.

2. Although a highly significant prolongation of survival was seen with the knockdown
and NK cells, none of the animals was cured of the tumor. This suggests that some
alternative mechanisms have been implemented so examination of these tumors, both
tumor cells and microenvironment would be important.

**Response:**

We sincerely thank the reviewer for this insightful comment. As the reviewer suggested,
we examined the levels of alternative factors in late-stage tumours from both the NSG
and C57BL/6 models (Supplementary Fig. 7). We used the reported human-specific
antigen TRA-1-85 (Zhai, K. et al., *Nat Cancer* 2021, doi:10.1038/s43018-021-00267-
9) to label GSCs isolated from the NSG model and measured some recognized NK cell-
suppressive factors, such as PD-L1, HLA-E and TGF- β 1. Among them, PD-L1 was

significantly upregulated under NK pressure (Supplementary Fig. 7a-b). In addition,
we also investigated microenvironmental factors in an immunocompetent mouse model
and found that large numbers of PD-L1-positive TAMs infiltrated the TME
(Supplementary Fig. 7e), which is similar to the results in a previous report that
established a GBM model in immunocompetent mice (Gangoso, E. et al., *Cell* 2021,
doi:10.1016/j.cell.2021.03.023). These new data suggested that GBM could evade NK
pressure by upregulating PD-L1/PD-1 pathway. Therefore, we administered the triple
combination of EZH-92aa knockdown, NK cell orthotopic injection and ICB therapy
in immunocompetent mouse models. As expected, a further synergistic effect was
observed on this model and prolonged experimental mice OS (2/5) to more than 70
71 days (Fig. 7f-i), indicating that EZH2-92aa inhibition-sensitized NK cell therapy may
work well with current ICB therapy.

3. The use of human GSCs is important because, as mentioned, these are the most
resistant component of the human cancer to treatment, but the mouse model requires
use of an immunocompromised host. Therefore, important interactions with other
cellular components of the immune system (ie T cells) are lacking. The investigators
may want to consider syngeneic models where comparable studies can be done in
immunocompetent animals.

Response:

We sincerely thank the reviewer for this excellent suggestion. In the revised manuscript,
we used C57BL/6 mice and the mouse glioma cell line GL261 to establish a comparable
model. CircEZH2 and its ORF manifest high conservation between humans and mice
(Supplementary Fig. 2a, b). We first validated the existence of mouse EZH-92aa in
GL261 cells (Supplementary Fig. 2e) and then verified the successful stable knockdown
of EZH2-92aa (Supplementary Fig. 2f). Next, we implanted GL261 control cells (stably
expressed scramble shRNA) or cells with EZH2-92aa stable KD into the brains of
C57BL/6 mice and administered the indicated treatment, as shown in Fig. 7f. Inspired
by the data from late-stage tumours described in above response, we combined ICB
therapy with EZH2-92aa inhibition-sensitized NK cell therapy in this model,

identifying a synergistic effect of these therapies (Fig. 7f-g).

4. Some minor points to consider:

a. Page 3, line 91: “Disruption of the ligand interactions... “ This is a major statement
and premise for this work, but is not referenced.

**Response:**

We sincerely apologize for this mistake. The reference (Eisele, G. et al. *Brain* 2006;
Crane, C. A. et al. *Neuro-Oncol* 2010) has been added to the revised manuscript.

b. Page 3, line 103” In the current study, we found that circular EZH2 (circEZH2)
expression is highly related to NK cell cytotoxicity.” This is also an important statement,
but it is not clear if there is a correlation or a cause and effect relationship. If known,
this should be clarified.

**Response:**

We sincerely thank the reviewer for this insightful comment. Based on the evidence
available, we postulate that circEZH2 plays a critical role in regulating NK cell
cytotoxicity. The reduced susceptibility to NK cell cytotoxicity of GBM cells might at
least partially be attributed to high expression of circEZH2/EZH-92aa in these cells. In
the revised manuscript, we have provided a clearer statement (“In the current study, we
found that circular EZH2 (circEZH2) plays a critical role in suppressing NK cell
cytotoxicity in GBM.”).

c. Page 5, line 143: “the expression of identified NK cell markers, including EGRs,
IFGN, TNF, and 1 GZMB (Barrow et al., 2018, Supplemental Table 4), was negatively
correlated with circEZH2 expression ($P < 0.01$)”. Since a series of NK cell markers were
evaluated, was a multiple comparisons correction used?

**Response:**

We sincerely thank the reviewer for this good question. The conclusion described above
was based on gene set enrichment analysis (GSEA) (Fig. 1f), which evaluates the
chosen markers as a whole (a gene set) instead of evaluating each single gene

individually. The expression of the gene set consisting of 27 chosen genes, as a whole,
was negatively correlated with circEZH2 expression. The listed genes were ranked by
Pearson's correlation coefficients. A detailed description of the GSEA method has been
added to the "Materials and methods" section. In this case, a multiple comparison
correction was not needed.

We also apologize for a mistake in the initial submitted manuscript: 'P<0.01', which
was inconsistent with the data shown in Fig. 1f. We have corrected this error in the
revised manuscript (P=0.012).

130 d. Page 5, line 169: "The above findings indicated that circEZH2 is a potential
oncogenic circRNA in GBM." Oncogenic implies that this gene is responsible for
causing the cancer, rather than helping malignant cells avoid immune recognition. An
alternative phrase should be considered.

**Response:**

We sincerely thank the reviewer for this good question. In the revised manuscript, we
have changed the term to "immunosuppressive" ("Based on these findings, circEZH2
is a potential upregulated immunosuppressive circRNA in GBM.").

e. Page 7, line 208: "Higher EZH2-92aa expression was negatively correlated with OS,
indicating that EZH2-92aa may be an independent prognostic marker in GBM". I
would suggest adding a qualifier that a multivariable analysis would need to be done
incorporating established prognostic factors to confirm that this is an independent
prognostic factor. For example, if EZH2-92aa overexpression occurs exclusively in
grade 4 IDH wildtype cancers and the data include IDH mutated grade 4 tumors, then
that may be sufficient to account for the survival difference.

**Response:**

We sincerely thank the reviewer for this excellent question. In the revised manuscript,
we collected the data from the chosen glioma cohort harbouring grade 4 IDH wildtype
GBM, grade 4 IDH mutated astrocytoma, as well as grade 3 1p19q co-deleted
oligodendrocytoma, and performed a multivariate Cox regression analysis. The results

(Supplementary Fig. 2g) indicated that EZH2-92aa, age, IDH1 mutation, and 1p19q
codeletion were all significant prognostic factors. We also performed a subgroup
analysis (Supplementary Fig. 2g, right panel) stratified by age, IDH1 mutation and
1p19q codeletion status to further evaluate whether EZH2-92aa was independent of age,
IDH1 mutation and 1p19q codeletion status. EZH2-92aa was independent of these three
factors, suggesting EZH2-92aa could be an independent prognostic marker for high-
grade gliomas.

*f. The final section talks about DDX2 and control of EZH2 expression. Although*
*interesting, it was not clear how this relates to the extensive discussion of the*
*mechanisms of NK cell inhibition with stabilization of EZH2, ligand suppression etc.*
*This section may be better placed earlier after the discussion of EZH2-92aa and EZH2*
*expression control and before the efficacy studies.*

**Response:**

We sincerely thank the reviewer for this good suggestion. In the revised manuscript, we
have rearranged the figures as suggested.

**Reviewer #2 (Remarks to the Author): with expertise in glioblastoma, NK cells**

*The study identified a new circular EZH2 RNA (circEZH2) to be expressed at high*
*levels in certain glioblastoma stem cells (GSCs) and confirmed circEZH2 translation*
*and expression of the respective novel EZH2-92aa protein. shRNA-mediated*
*knockdown of EZH2-92aa expression enhanced sensitivity of GSCs to the NK cell line*
*NK-92MI, which was correlated to a direct repressive effect of EZH2-92aa on MICA*
*and MICB transcription and an indirect effect on the expression of other NKG2D*
*ligands. While it is intriguing to assume reduced MICA/B expression and an ensuing*
*lack of NKG2D activation in NK cells as the underlying mechanism for EZH2-92aa-*
*induced escape of GBM cells from NK cell cytotoxicity, this appears to be an over-*
*interpretation of the actual data which lack formal proof for the importance of the*
*MICA/B-NKG2D interaction and respective confirmation with primary human NK cells.*

Major:

1. sh-circEZH2 mediated knockdown of EZH2-92aa expression reduced MICA/B mRNA
levels in MES28 cells (Figure 4B). Data confirming that this is also the case for surface
expression of MICA/B proteins are missing.

**Response:**

We sincerely apologize for failing to include these data. In the revised manuscript, we
used WB and flow cytometry to validate the increased MICA/B levels after EZH2-92aa
knockdown in MES28 and GSC23 cells, and the results were consistent with the qPCR
data (Fig. 5b and Fig. S4d, f).

2. Direct proof for a major role of the MICA/B-NKG2D axis for enhanced NK cell
cytotoxicity against knockdown cells is lacking. Cytotoxicity experiments with wildtype
and EZH2-92aa knockdown GSCs (increased MICA/B mRNA) and NK cells in the
presence of a blocking NKG2D antibody or NKG2D-Fc should be included to clarify
to what extent activating receptors other than NKG2D are possibly involved, and
whether enhanced NK sensitivity in the knockdown cells is indeed mainly due to
restored MICA/B-NKG2D interactions.

**Response:**

We sincerely thank the reviewer for these excellent suggestions. Accordingly, we added
a blocking NKG2D antibody (10 µg/ml) control during the co-culture of EZH2-92aa
KD GSCs and NK cells (Fig. 6i, j and Supplementary Fig. 5b-d). While EZH2-92aa
knockdown resulted in increased NK cell cytotoxicity towards GSCs, the addition of
NKG2D blocking antibody reversed this effect, suggesting that EZH2-92aa regulates
GSC sensitivity to NK cells mainly through the NKG2D-NKG2DL axis.

3. In the absence of data on surface MICA/B protein expression and functional
experiments clearly demonstrating reduced MICA/B-NKG2D interaction as the cause
for NK resistance, the title of the manuscript, parts of abstract and discussion, as well
as the proposed model in Figure 7F are overstating what is actually shown in the study.

**Response:**

We sincerely apologize for the missing data. In this revised manuscript, we added
supporting evidence based on the reviewers' kindly suggestions, including increased
MICA/B protein expression after EZH2-92aa KD (Fig. 5b and Supplementary Fig. 4d,
f) and NK cytotoxicity assays with NKG2D blockade (Fig. 6i, j and Supplementary Fig.
5b-d). We believe the current evidence better fits the statement and conclusion in the
revised manuscript.

4. Alternative triggers of NK cell activity are downregulation of MHC class I and NCR
activation, which may also be affected by circEZH2/EZH2-92aa. Indeed, EZH2-92aa
knockdown in MES28 cells appears to have resulted in increased NCR3LG1 (=B7H6)
mRNA expression (Figure 3A), which is a ligand for NKp30. The authors should
comment on possible direct or indirect effects of EZH2-92aa on NCR ligand expression.

Did EZH2-92aa knockdown have any effect on MHC class I expression?

**Response:**

We sincerely thank the reviewer for this good question. In this revised manuscript, we
evaluated the effect of EZH2-92aa on NCR ligand expression by qPCR, including
vimentin (ligand for NKp46), MLL5, PCNA, NID1, PDGF-D (ligands for NKp44),
Galectin-3, and B7-H6 (ligands for NKp30) (Chen, J. et al., *Front. Oncol.* 2020, doi:
10.3389/fonc.2020.00874). None of these ligands showed alterations after
circEZH2/EZH2-92aa knockdown (Supplementary Fig. 4b). Specifically, although B7-
H6 (NCR3LG1) exhibited upregulation in the GSEA drawn from RNA-seq data in Fig.
4a (in the revised manuscript), its expression was validated to be unchanged upon
circEZH2 KD in the qPCR experiments (Supplementary Fig. 4b).

We also measured MHC-I levels in EZH2-92aa stable KD cells with verified qPCR
primers (Dersh, D. et al. *Immunity* 2021, doi: 10.1016/j.immuni.2020.11.002) and a pan
MHC-I antibody. MHC-I levels were unaffected by EZH2-92aa knockdown
(Supplementary Fig. 4c). Our data collectively indicating that EZH2-92aa perturbs NK
activation by interfering with the NKG2DL-NKG2D axis.

5. NK cell cytotoxicity in vitro and in vivo was investigated using the NK-92MI cell line

as a model. NK-92 cells are known for their lack of most of the inhibitory NK cell
receptors and their high intrinsic cytotoxicity, which distinguishes them from primary
NK cells expanded from PBMCs. In vitro cytotoxicity experiments with wildtype and
EZH2-92aa knockdown GSCs should also be performed with ex vivo expanded primary
NK cells from different donors to investigate whether EZH2-92aa-mediated NK
resistance is indeed a general effect or limited to the particular NK cell line used.

**Response:**

We sincerely thank the reviewer for this good suggestion. Using previously reported
protocols (Zheng, X. et al., *Nat Immunol* 2019, doi: 10.1038/s41590-019-0511-1), we
sorted and expanded NK cells from the peripheral blood of two human donors (NK #1
and NK#2). We repeated the in vitro cytotoxicity experiments (including LDH and
calcein AM cytotoxicity assays) with these cells. Similar results were obtained
compared to the NK-92MI cell line, as shown in Supplementary Fig. 3c-e, implying
that EZH2-92aa-mediated NK resistance is likely to be a general effect.

6. The data on the interaction of ectopically expressed EZH2-92aa-3xflag or the
overexpressed EZH2-92aa open reading frame with MICA and MICB promoter regions
and their inhibitory effects on the expression of MICA/B luciferase reporter constructs
are convincing, but need to be confirmed by respective experiments demonstrating
similar effects for endogenous EZH2-92aa. The EMSA experiments shown in Figure 4I
and 4J should be repeated with nuclear extracts of wildtype MES28 and GSC23 cells
which already have high endogenous levels of EZH2-92aa protein as shown in Figure
2E. Rabbit anti-EZH2-92aa antibody may be used for supershift experiments.
Alternatively, ChIP experiments could be performed with endogenous EZH2-92aa and
the EZH2-92aa antibody (Figure 4G and 4H).

**Response:**

We sincerely thank the reviewer for these excellent suggestions. The EMSA experiment
in Fig. 5i and j (in the revised manuscript) was repeated using MES28 and GSC23
nuclear extracts (Supplementary Fig. 4j). ChIP experiments were also performed with
the EZH2-92aa antibody (Supplementary Fig. 4h), showing endogenous binding of

EZH2-92aa to MICA/B promoters. Unfortunately, we noted limitations with the use of
the EZH2-92aa antibody in supershift experiments. Although we have tried many times,
the supershift band was not observed. We propose that the current data, which include
ChIP and EMSAs using endogenous EZH2-92aa and Flag-tag antibodies, supershift
experiments with the Flag-tag antibody and assays with mutated EMSA probes, are
sufficient to show the role of EZH2-92aa in directly regulating the transcriptional
activity of MICA/B.

7. In a previous study cited by the authors expression of FBXW7-185aa, the product of
a circular FBXW7 RNA, was found to inhibit proliferation of GBM cells (Yang et al.
2018). Was reduced circ-FBXW7 found in GSCs with upregulated EZH2-92aa? The
authors may also want to comment on how circ-FBXW7 would fit into their hypothesis
that EZH2-92aa protects EZH2 protein from FBXW7-induced degradation.

**Response:**

We sincerely thank the reviewer for this good question. In our previous paper,
circFBXW7 was reported to be a tumour-suppressive circRNA. Its coding product
FBXW7-185aa binds to deubiquitinase USP28, preventing USP28 from binding to
FBXW7 and thus liberating FBXW7 to degrade the oncoprotein c-myc. In this paper,
EZH2-92aa interacted with FBXW7, functioned as a decoy itself to reduce the ubiquitin
level on EZH2 and protected it from degradation. We investigated the PubMed database
and found that USP28 has not been reported to stabilize EZH2 thus far. Therefore,
EZH2-92aa may not be directly associated with circFBXW7 and FBXW7-185aa in
regard to the mechanism. We measured the level of circFBXW7 in GSCs using qPCR
(data are described below) and found that it was downregulated in these cell lines
(consistent with its tumour-suppressive role), but no significant correlation was
observed between circFBXW7 and circEZH2 expression (Fig. 1j) in GSC cell lines.
Based on these data, we suppose that circEZH2-92aa independently exerted its
biological function from circFBXW7-185aa.

8. EZH2-92aa knockdown on its own transiently reduced growth of brain tumor
 xenografts in mice, attributed by the authors to an adaptive mechanism of EZH2
 degradation-induced H3K27me3 reduction. A more simple explanation would be that
 the growth disadvantage of EZH2-92aa knockdown cells favors the selective outgrowth
 of a subpopulation with less complete knockdown or loss of the shRNA construct.
 Accordingly, an experiment is missing comparing EZH2-92aa expression in knockdown
 cells from tissue culture and tumor cells growing out in the respective in vivo experiment.
 Did the authors confirm continued EZH2-92aa knockdown in late passage GSCs that
 resumed normal in vitro growth characteristics (Figure S5)?

**Response:**

We sincerely thank the reviewer for this good question and apologize for failing to
 include the data in the first submitted manuscript. In this revision, we detected the
 expression of EZH2-92aa in late-passage GSCs and in late-stage xenografts from *in*
 *vivo* models again to validate the effective and continued knockdown of EZH2-92aa in
 these samples. We collected and digested the xenografts, filtered and washed the
 suspension, and cultured the cells in GSC-specific medium for another 3 days before
 the detection of EZH2-92aa in GSCs from *in vivo* models. The results presented in
 Supplementary Fig. 6c-d show that effective knockdown of EZH2-92aa still occurred
 in late-passage cells or late-stage tumours. Therefore, we speculate that the resumed
 growth rate is probably due to the previously reported adaptive mechanism of EZH2
 degradation-induced H3K27me3 reduction.

9. In the last part of the study the authors show an RNA immunoprecipitation
experiment indicating direct interaction of DDX3 with circEZH2 RNA and demonstrate
reduced EZH2-92aa protein levels upon DDX3 knockdown, concluding from this a
direct involvement of DDX3 in the regulation of circEZH2 RNA translation. This should
be worded more carefully since DDX3 is involved in many different cellular processes,
and its knockdown may also have global effects that indirectly inhibit EZH2-92aa
protein expression. Along those lines, it appears that at least in GSC23 cells DDX3
knockdown also reduced expression of beta-actin (Figure 7C). An experiment
demonstrating enhanced MICA/B expression and NK sensitivity upon DDX3
knockdown as expected with the ensuing reduction in EZH2-92aa would connect this
part of the study better to the rest of it.

**Response:**

We sincerely thank the reviewer for this good question. In the revised manuscript, we
modified the original statement. We apologize for the data presented in Fig. 7c of the
initial submission. We believe that the reduced expression of beta-actin was due to
unequal loading of samples, which might be attributed to some inaccuracy when we
determined the protein concentration during sample preparation. Therefore, we
repeated this experiment and ensured that we loaded equal weights and volumes of
proteins into each lane, and the results showed that EZH2-92aa was downregulated
upon DDX3 knockdown, with the loading control present at the same levels across each
lane (Fig. 3c in the revised manuscript). Under the reviewer's suggestion, we also
observed increased NK cell sensitivity and MICA/B expression upon DDX3
knockdown (Supplementary Fig. 3i-1, 4g). Although DDX3 could have global effects
that indirectly reduced EZH2-92aa, our current data strongly implied their close
connection.

**Minor:**

1. References should be checked for accuracy. Some references indicated with a single
author are incomplete (see for example Eisele 2006).

**Response:**

We sincerely apologize for this mistake. We have checked the reference format again
in the revised manuscript and have ensured that the format complies with the journal's
guidelines.

2. In text citations should be adjusted to journal style. Some appear to refer to a certain
page number.

**Response:**

We sincerely thank the reviewer for this good suggestion. The revised manuscript has
been corrected strictly according to the journal's formatting instructions.

3. Line 25: Typing error: "vin".

**Response:**

We sincerely apologize for this mistake. We have corrected this error in the revised
manuscript.

4. Lines 347 to 348: "...resistance to GSCs 'induced' cellular toxicity". Was 'reduced'
meant?

**Response:**

We sincerely apologize for this mistake. We meant "reduced" indeed. This error has
been corrected in the revised manuscript.

**Reviewer #3 (Remarks to the Author): with expertise in circRNA (bioinformatics)**

In this manuscript, the authors characterized EZH2-92aa, a novel protein encoded 44
by circular EZH2 (circEZH2), in GBM, and demonstrated that EZH2-92aa induces the
45 resistance of glioblastoma stem cells (GSCs) to NK cells. They further showed that
stable EZH2-92aa knockdown enhanced NK cell-mediated eradication of 52 GSCs both
in vitro and in vivo. The manuscript is overall well-organized, and the experimental
data is solid and convincing. The reviewer has several comments for the authors to

further improve the manuscript:

1. Detailed information for Bioinformatics analysis is needed. For example, what is the
accession number for the RNA-seq data generated in this manuscript – it seems that
the link provided [<https://www.ncbi.nlm.nih.gov/sra/?term=SRP095744>] included only
two samples, and it is related to FBXW7 circRNA, not the current ones. How did they
quantify circRNAs (with which algorithm)? How do they identify differentially
expressed circRNAs (Figure 1A)? How did they quantify the expression of circEZH2 in
the clinical patient cohort (Figure 1J)? Several other bioinformatics analyses are
lacking the details which need to be specified.

**Response:**

We sincerely thank the reviewer for these good questions.

(1) We sincerely apologize for this error in the accession number. In the revised
manuscript, it has been corrected to PRJNA525736.

(2) The detailed methods or algorithms used to quantify circRNAs have been
supplemented in the revised manuscript (in the “Bioinformatics analysis” section).

(3) Glioma samples collected from patients were rapidly frozen in liquid nitrogen and
ground in liquid nitrogen before lysis to quantify the expression of circEZH2 in the
clinical patient cohort (Fig. 1k, l in the revised manuscript). RNA extraction and
reverse transcription were performed according to the manufacturers’ instructions.
Then, we applied qPCR to quantify the relative circEZH2 levels compared to
peritumour control samples.

(4) The details of other bioinformatics analyses, such as GSEA, have also been added
to the revised “Methods and materials” section.

2. How do they characterize the length of circRNAs (Figure 1B)? How to interpret the
association between circEZH2 and NK markers (Figure 1F), since many markers are
actually negatively correlated with circEZH2. Furthermore, did the authors check the
associations between circEZH2 with other immune cells, for example, T cells. Is it
possible for circEZH2 to exert functions through T cells?

**Response:**

We sincerely thank the reviewer for these good questions.

(1) Fig. 1b describes the length distribution of ORFs from identified circRNAs. We
 used a reported method (Pamudurti, N. R. et al., *Molecular Cell* 2017, doi:
 10.1016/j.molcel.2017.02.021) to annotate the most likely circORF and calculate
 its length. The details have been updated in the revised “Methods and materials”
 section.

(2) Fig. 1f shows that the gene set consisting of 27 chosen marker genes negatively
 correlated with circEZH2 expression as a whole. The detailed method is described
 below. We used the expression level of circEZH2 as a molecular phenotype and
 calculated Pearson’s correlation coefficients between the expression of circEZH2
 and each protein-coding gene. Next, the coding genes were ranked by Pearson’s
 correlation coefficients, and GSEA was performed to show that circEZH2
 expression was negatively correlated with the expression of the selected gene set.
 Indeed, as shown in the heatmap in the right panel of Fig. 1f, more than half of the
 chosen markers were negatively correlated with circEZH2. Therefore, these
 markers are generally in a relatively low position in the GSEA ranking (according
 to the value of the correlation coefficient). Accordingly, we refined the legend for
 Fig. 1f in this revised version.

(3) We assessed the association between circEZH2 and other immune gene sets in the
 KEGG and GO_immue datasets. The significant results are presented in
 Supplementary Table 3. The enriched T-cell-related pathways failed to show a
 significant p value (>0.05), which is not included in the supplementary table. Some
 of the results for T cells are listed below, with p values showing nonsignificant
 differences (we would be happy to share the full source file upon request).

	GS follow link to MSigDB	SIZE	ES	NES	NOM p-val	FDR q-val	FWER p-val	RANK AT MA	LEADING EDGE
19	GO_NEGATIVE_REGULATION_OF_T_CELL_MEDIATED_IMMUNITY	19	-0.49	-1.21	0.214	1	1	4233	tags=42%, list=22%, signal=54%
30	GO_T_CELL_ACTIVATION_INVOLVED_IN_IMMUNE_RESPONSE	102	-0.27	-0.93	0.617	1	1	4236	tags=24%, list=22%, signal=30%
33	GO_T_CELL_DIFFERENTIATION_INVOLVED_IN_IMMUNE_RESPONSE	68	-0.28	-0.9	0.704	1	1	4139	tags=21%, list=22%, signal=26%
52	GO_ALPHA_BETA_T_CELL_ACTIVATION_INVOLVED_IN_IMMUNE_RESPONSE	60	-0.23	-0.73	0.964	1	1	4139	tags=18%, list=22%, signal=23%
54	GO_T_CELL_MEDIATED_IMMUNITY	101	-0.2	-0.69	0.995	1	1	4233	tags=20%, list=22%, signal=25%

	GS follow link to MSigDB	GS DETAIL	SIZE	ES	NES	NOM p-val	FDR q-val	FWER p-val	RANK AT MA	LEADING EDGE
77	KEGG_T_CELL_RECEPTOR_SIGNALING_PATHWAY		107	-0.25	-0.87	0.79	0.888	1	3997	tags=22%, list=21%, signal=28%

3. The authors showed the consistency between circEZH2 with circBase data – which
 is actually not a cancer circRNA database. Did the authors compare with other data
 resources, such as MiOncoCirc, CircRic, CSCD/CSCD2?

**Response:**

We sincerely thank the reviewer for these good questions.

The ID for circEZH2 in circBase is *hsa_circ_0006357*. Some key information is listed
 as follows: genomic position: chr7:148,543,561-148,544,397; genomic length: 836 bp;
 and spliced length: 253 bp.

According to the reviewer’s suggestion, we searched the database for circEZH2 using
 the key characteristics described above.

(1) MiOncoCirc

CircEZH2 does not appear to be included in the MiOncoCirc online database.

(2) CircRic (<https://hanlab.uth.edu/cRic/expr/>)

CircEZH2 was found in this database, ranking first. It has been reported in three
 cancer cell lines: LCLL, LGG and LUSC.

CircRNA	No. of Cancer Cell Lines	Average Backsplicing Reads Count	No. of Cancer Lineages	Plot
7_148543561_148544397 EZH2	3	2	3	
7_148514109_148514313 EZH2	2	2	2	
7_148524255_148544397 EZH2	1	44	1	
7_148516687_148544397 EZH2	1	10	1	
7_148511177_148511315 EZH2	1	3	1	
7_148516212_148516648 EZH2	1	2	1	
7_148514968_148516779 EZH2	1	2	1	
7_148506028_148506162 EZH2	1	2	1	
7_148504867_148506162 EZH2	1	2	1	
7_148514215_148514432 EZH2	1	2	1	

(3) CSCD2 (<http://geneyun.net/CSCD2/>)

Due to the use of a different version of the genome, the exact genomic position of
 circEZH2 in this database is chr7:148846469|148847305 (different from that in
 circBase). However, the genomic length and spliced length are identical to those in
 circBase.

chr7:148832633 148847305	EZH2	UCSC	normal,cancer	hypopharyngeal_cancer_adj_normal_2,Breast_cancer_adj_n
chr7:148836821 148847305	EZH2	UCSC	normal,cancer	colon_carcinoma_8,pulmonary_artery_endothelial_cell_2,cc
chr7:148846469 148847305	EZH2	UCSC	normal,cancer	Breast_cancer_adj_normal_11,placental_epithelial_cell_1,IM
chr7:148846496 148847305	EZH2	UCSC	normal,cancer	A549_1,hypopharyngeal_cancer_adj_normal_3,fibroblast_of
chr7:148819736 148826295	EZH2	UCSC	normal,cancer	K562_4,hair_follicle_dermal_papilla_cell_2,fibroblast_of_de
chr7:148810735 148826108	EZH2	UCSC	normal,cancer	colson_1,Tongue_squamous_cell_carcinoma_adj_normal_1

Overview	CircRNA	MRE	RBP	ORF	Full length
Sequence					
<pre>>sequence AATAATCATGGGCCAGACTGGAAGAAATCTGAGAAGGGACCAGTTTGTGGCGGAAGCGTGAAAATCA GAGTACATCGGACTGAGACAGCTCAAGAGGTTGACACGAGCTGATGAAGTAAAGAGTATGTTTAGTTCCA ATCGTCAGAAAATTTGGAAAGAACGGAAATCTTAAACCAAGAATGGAAACAGCGAAGGATACAGCCTGT GCACATCCTGACTTCTGTGAGCTCATTGCGCGGGACTAGGGAG</pre>					

4. It is interesting to see the inhibition of EZH2-92aa sensitizes GSCs to NK cell
cytotoxicity. However, the mechanism for this section need further efforts to make it
clear.

**Response:**

We sincerely thank the reviewer for this good question. In the revised manuscript, we
added evidence based on the reviewers' kind suggestions, including increased MICA/B
protein expression after EZH2-92aa KD (Fig. 5b and Supplementary Fig. 4d, f), NK
cytotoxicity assays with NKG2D blockade (Fig. 6i, j and Supplementary Fig. 5b-d), the
investigation of other factors involved in NK activation (NCR ligands, etc.,
Supplementary Fig. 4a-c) and ChIP and EMSAs using endogenous EZH2-92aa
antibodies (Supplementary Fig. 4h, j). We believe the current evidence better fits the
statement and conclusion in the revised manuscript.

**Reviewer #4 (Remarks to the Author): with expertise in circRNA**

This study identified a circular RNA (circEZH2) that was upregulated in Glioblastoma
(GBM). circEZH2 was found to encode a polypeptide EZH2-92aa. The authors
provided a series of data to support the coding ability of circEZH2 and the real presence
of EZH2-92aa. Further characterizations of EZH2-92aa demonstrated that EZH2-92aa
mediated NK cell tolerance of GBM through transcriptional regulation of MICA/B

expression and posttranscriptional regulation of EZH protein degradation. Analyses of
human samples and public data along with mice models were also used to demonstrate
effects of EZH2-92aa in GBM immune evasion.

Generally this study is very well designed and the data are compelling to make the
conclusions. I only have three minor concerns.

1, For the part of EZH2-92aa suppresses the transcription of MICA/B, it is better to
perform some nuclear run-on assays, under EZH2-92aa overexpression and
knockdown conditions, respectively.

**Response:**

We sincerely thank the reviewer for this good suggestion. The nuclear run-on assay
showing the effect of EZH2-92aa on suppressing MICA/B nascent transcription has
been added to the revised manuscript (Supplementary Fig. 4e), and we used an updated
version of previously reported protocols (Roberts, T. C. et al., *Nature Protocols*, 2015,
doi: 10.1038/nprot.2015.076).

2, Please provide more details about circEZH2. Information about: conservancy (mice
circEZH2, existence or coding potency?), the possible presence of Exon-intron circRNA
(EIciRNA) isoform of circEZH2 (and if there is EIciEZH2, please at least discuss its
potential role in transcriptional regulation), how about the ENNHGPDWEEI epitope
(is it unique in sequences in human?), how about functional domain or motif in EZH2-
92aa? How about the copy numbers per cell of circEZH2?

**Response:**

We sincerely thank the reviewer for these good questions.

(1) Conservancy

Murine circEZH2 has been identified (circBase ID: mmu_circ_0001471) and
manifests a highly conserved sequence, ORF and coding ability (Supplementary
Fig. 2a, b, e, f) to human circEZH2.

(2) EICIcRNA

We employed a database of full-length circRNA sequences – circAtlas
(<http://circatlas.biols.ac.cn>) – to address this issue. This database integrates over
one thousand RNA-seq samples from 19 tissues across six vertebrate species and
leverages four different algorithms to recognize circRNAs in each sample. We
searched for all the circRNAs derived from the parent gene EZH2
(ENSG00000106462) and obtained 80 entries. The first circRNA (circAtlas ID:
hsa-EZH2_0001, spliced length 253 nt) is the exact circRNA we studied in this
paper. The circRNA type of the 80 identified entries was categorized into four
groups: antisense, exon, intron, and nonrepeat. EICIcRNA of EZH2 has not been
discovered yet, based on current sequencing data.

(3) NNHGPDWEEI epitope

We used the blastp tool on the NCBI website and searched for potential proteins
containing this sequence in the nonredundant protein sequence and reference
protein databases. No protein included in these databases showed a completely
identical sequence to the unique 10 aa C-terminus of EZH2-92aa. To our current
knowledge, this epitope is unique in humans.

(4) Domain and motif

First, we applied the Uniprot database to compare the protein sequences between
EZH2-92aa and EZH2 (ID: Q15910). The shared sequences are located at the N-
terminus, while some well-known annotated domains of EZH2 (CXC, SET) are not
included in this region. However, there may be some shared points involved in
interaction with other proteins as shown below:

We also employed the MEME tool to predict potential domains or motifs in EZH2-

92aa. The results are shown below. It seems that the function of these potential
 domains need investigation in the future study, which we are currently working on.

 (5) Copy numbers of circEZH2

We performed absolute quantification of circEZH2 in GSC and NHA cell lines by
 qPCR, following previous reported methods (Liu, C.-X. *et al. Cell* 2019, doi:
 10.1016/j.cell.2019.03.046). The results are shown below (ranging from 2-12 copies
 548 per cell in these cell lines):

 3, The authors made a strong argument about roles of EZH2-92aa, although it would
 not be appropriate to just assume that circEZH2 only serves as the translational
 template for EZH2-92aa, without any other role as a circRNA (or even the generation
 and metabolism of this RNA may play some role in GBM). Please discuss this kind of

*possibility and tune down the significance of EZH2-92aa (such as “provided a*
*potentially important combination strategy for GBM intervention”).*

**Response:**

We sincerely thank the reviewer for this good suggestion. In this study, we distinguished
whether circEZH2 or EZH2-92aa inhibited NK cell cytotoxicity by overexpressing
constructs, including mut (overexpression of mutated circEZH2 with deletion of ATG),
circ (circEZH2) and ORF (linearized circEZH2 ORF), in GSC456 cells, as shown in
Fig. 4b and Supplementary Fig. 3f-h. Stable OV of the mutated construct, which lacks
the protein-coding ability, failed to enhance the resistance of GSC456 cells to NK cells.
The results suggested that, at least in the mechanism model of this paper, EZH2-92aa
exerts the main inhibitory effect on the NK cell-mediated immune response rather than
circEZH2. However, whether circEZH2 plays an important role in other biological is
an interesting issue that remains to be addressed in future studies processes. We have
corrected some statements in the revised manuscript according to the reviewer’s
suggestions.

REVIEWERS' COMMENTS

Reviewer #1 (Remarks to the Author):

The authors have done a very good job addressing my comments and concerns from the first version of the manuscript. I have no additional comments.

Reviewer #2 (Remarks to the Author):

New data have been added to the revised manuscript, improving the study and adequately addressing my prior questions and concerns.

Reviewer #3 (Remarks to the Author):

The authors nicely addressed the comments, and I have one minor comment - that the authors may cite those papers mentioned in the response letter.

Reviewer #4 (Remarks to the Author):

All my concerns are successfully addressed.

REVIEWERS' COMMENTS (for the revised manuscript)

Reviewer #1 (Remarks to the Author):

The authors have done a very good job addressing my comments and concerns from the first version of the manuscript. I have no additional comments.

Reviewer #2 (Remarks to the Author):

New data have been added to the revised manuscript, improving the study and adequately addressing my prior questions and concerns.

Reviewer #3 (Remarks to the Author):

The authors nicely addressed the comments, and I have one minor comment - that the authors may cite those papers mentioned in the response letter.

Response:

Thanks for the suggestion. The citations have been added in the manuscript.

Reviewer #4 (Remarks to the Author):

All my concerns are successfully addressed.